J Physiol 603.13 (2025) pp 3837–3856

# Daily blood flow restriction does not preserve muscle mass and strength during 2 weeks of bed rest

Cas J. Fuchs[1] , Wesley J. H. Hermans[1], Jean Nyakayiru[1], Michelle E. G. Weijzen[1], Joey S. J. Smeets[1], Thorben Aussieker[1], Joan M. Senden[1], Will K. H. W. Wodzig[2], Tim Snijders[1], Lex B. Verdijk[1] and Luc J. C. van Loon[1]

[1]*Department of Human Biology, Faculty of Health, Medicine and Life Sciences, School of Nutrition and Translational Research in Metabolism (NUTRIM), Maastricht University Medical Centre+, Maastricht, The Netherlands*
[2]*Central Diagnostic Laboratory, Maastricht University Medical Centre+, Maastricht, The Netherlands*

Handling Editors: Paul Greenhaff & Matthew Brook

The peer review history is available in the Supporting Information section of this article (https://doi.org/10.1113/JP286065#support-information-section).

**Abstract** We measured the impact of blood flow restriction on muscle protein synthesis rates, muscle mass and strength during 2 weeks of strict bed rest. Twelve healthy, male adults (age: $24 \pm 3$ years, body mass index: $23.7 \pm 3.1$ kg/m$^2$) were subjected to 14 days of strict bed rest with unilateral blood flow restriction performed three times daily in three 5 min cycles (200 mmHg). Participants consumed deuterium oxide and we collected blood and saliva samples throughout 2 weeks of bed rest. Before and immediately after bed rest, lean body mass (dual-energy X-ray absorptiometry scan) and thigh muscle volume (magnetic resonance imaging scan) were assessed in both the blood flow restricted (BFR) and control (CON) leg. Muscle biopsies were collected and unilateral muscle strength (one-repetition maximum; 1RM) was assessed for both legs before and after the bed rest period. Bed rest resulted in $1.8 \pm 1.0$ kg lean body mass loss ($P < 0.001$). Thigh

**Cas Fuchs** is a post-doctoral researcher and teacher at Maastricht University Medical Centre+ (MUMC+), in the Department of Human Biology. He obtained his PhD from Maastricht University, where he investigated strategies for post-exercise recovery, with a specific emphasis on carbohydrate and protein metabolism as well as cooling and heating interventions. Currently, he focuses his research on exercise and nutrition metabolism as well as body composition, utilizing advanced methodologies such as stable isotope tracer methodology and magnetic resonance imaging/spectroscopy.

The Journal of Physiology

muscle volume declined from 7.1 ± 1.1 to 6.7 ± 1.0 L in CON and from 7.0 ± 1.1 to 6.7 ± 1.0 L in BFR ($P < 0.001$), with no differences between treatments ($P = 0.497$). In addition, 1RM leg extension strength decreased from 60.2 ± 10.6 to 54.8 ± 10.9 kg in CON and from 59.2 ± 12.1 to 52.9 ± 12.0 kg in BFR ($P = 0.014$), with no differences between treatments ($P = 0.594$). Muscle protein synthesis rates during bed rest did not differ between the BFR and CON leg (1.11 ± 0.12 *vs*. 1.08 ± 0.13%/day, respectively; $P = 0.302$). Two weeks of bed rest substantially reduces skeletal muscle mass and strength. Blood flow restriction during bed rest does not modulate daily muscle protein synthesis rates and does not preserve muscle mass or strength.

(Received 29 November 2023; accepted after revision 8 February 2024; first published online 26 February 2024)

**Corresponding author** L. J. C. van Loon: Department of Human Biology, Faculty of Health, Medicine and Life Sciences, School of Nutrition and Translational Research in Metabolism (NUTRIM), Maastricht University Medical Centre+, PO Box 616, 6200 MD Maastricht, the Netherlands.     Email: l.vanloon@maastrichtuniversity.nl

**Abstract figure legend** Two weeks of bed rest lowers skeletal muscle mass and strength. Daily passive blood flow restriction during bed rest does not modulate daily muscle protein synthesis and does not preserve skeletal muscle mass or strength. Created with BioRender.com.

## Key points

- Bed rest, often necessary for recovery from illness or injury, leads to the loss of muscle mass and strength. It has been postulated that blood flow restriction may attenuate the loss of muscle mass and strength during bed rest.
- We investigated the effect of blood flow restriction on muscle protein synthesis rates, muscle mass and strength during 2 weeks of strict bed rest.
- Blood flow restriction applied during bed rest does not modulate daily muscle protein synthesis rates and does not preserve muscle mass or strength.
- Blood flow restriction is not effective in preventing muscle atrophy during a prolonged period of bed rest.

## Introduction

Recovery from injury or illness often necessitates a prolonged period of bed rest. This reduces insulin sensitivity, decreases muscle oxidative capacity, lowers muscle protein synthesis rates, and leads to a progressive loss of skeletal muscle mass and strength in humans of all ages (Dirks et al., 2016; English et al., 2016; Stuart et al., 1988; Wall & van Loon, 2013). Bed rest has been shown to reduce skeletal muscle mass by ∼0.5% per day, with a concomitant decline in strength of ∼0.3–4.2% per day (Wall & van Loon, 2013). It has been suggested that the impact of short, successive periods of disuse may be largely responsible for the loss of muscle mass and strength that are typically observed throughout lifespan (English & Paddon-Jones, 2010; Wall & van Loon, 2013). As a result, it is of major clinical importance to find effective strategies to preserve muscle mass quantity and quality during whole-body disuse.

Blood flow restriction has recently been proposed as an effective strategy to attenuate the loss of muscle mass and strength during a period of skeletal muscle disuse (Patterson et al., 2019). With blood flow restriction an external pressure device is applied to reduce arterial blood inflow and restrict venous outflow in the upper or lower limbs. This results in reduced oxygen supply and venous blood pooling within the capillary network (Patterson et al., 2019), which may lead to skeletal muscle tissue hypoxia, increase metabolic stress and cause cell swelling. This may stimulate angiogenesis and augment muscle oxidative capacity (Jeffries et al., 2018; Li et al., 2022; Pignanelli et al., 2021). These mechanisms have also been implicated in promoting skeletal muscle anabolism (Loenneke et al., 2010, 2012b,c; Martin et al., 2022). Blood flow restriction has been applied in combination with (low-intensity) exercise to effectively augment muscle protein synthesis rates (Fry et al., 2010; Fujita et al., 2007; Gundermann et al., 2014; Nyakayiru et al., 2019) and, as such, to increase muscle mass and strength gains during more prolonged exercise training (Patterson et al., 2019; Pearson & Hussain, 2015; Wernbom et al., 2008). Blood flow restriction has also been proposed to preserve muscle mass during muscle disuse (Loenneke et al., 2012a; Martin et al., 2022; Patterson et al., 2019). Whereas some

studies (Kakehi et al., 2020; Kubota et al., 2008, 2011; Takarada et al., 2000) have reported blood flow restriction to attenuate the decline in muscle cross-sectional area and strength, other studies have failed to mitigate the loss of muscle mass and strength following ∼2 weeks of limb immobilization (Slysz et al., 2020) or recovery from anterior cruciate ligament reconstruction (Iversen et al., 2016). Consequently, the efficacy of blood flow restriction as a means to preserve muscle mass and strength during disuse remains to be consistently established.

Whereas previous studies have utilized limb immobilization as the model of disuse, no studies have investigated the effects of daily passive blood flow restriction on skeletal muscle mass and strength during a prolonged period of bed rest. In addition, it remains largely unknown what the potential mechanisms may be by which blood flow restriction could attenuate the loss of muscle mass and strength during disuse. In the current study we subjected 12 healthy young males to 2 weeks of strict bed rest and applied daily blood flow restriction to one leg. We used comprehensive measures of skeletal muscle mass and strength in combination with skeletal muscle biopsies and a stable isotope tracer methodology to investigate potential mechanisms by which blood flow restriction could impact muscle atrophy during bed rest. Given that many studies have shown blood flow restriction to mitigate the decline in both muscle quantity and quality during limb immobilization, we hypothesized that daily blood flow restriction would attenuate the decline in muscle oxidative capacity, activate anabolic signalling pathways, increase muscle protein synthesis rates and, as such, attenuate the loss of skeletal muscle mass and strength during 2 weeks of bed rest.

## Methods

### Subjects

Twelve healthy young men (age $24 \pm 3$ years) were included in the current study. Subjects' characteristics are presented in Table 1. All subjects were informed on the nature and risks of the experiment before written informed consent was obtained. The current study was approved by the Medical Ethical Committee of Maastricht University Medical Centre+ (registration number 17-3-014) in accordance with the *Declaration of Helsinki*. This trial was registered at the former Dutch trialregister.nl with the number NL6222. This is now available via https://www.onderzoekmetmensen.nl/en/trial/22281.

### General study design

The experimental design and procedures are displayed in Table 2. After a screening session and inclusion into

**Table 1. Baseline subjects' characteristics**

|  | Subjects ($n = 12$) |
| --- | --- |
| Age (years) | $24 \pm 3$ |
| Body mass (kg) | $79.4 \pm 12.4$ |
| Height (m) | $1.83 \pm 0.07$ |
| BMI (kg/m$^2$) | $23.7 \pm 3.1$ |
| Fasting glucose (mmol/L) | $5.4 \pm 0.6$ |
| Fasting insulin (mU/L) | $9.8 \pm 5.4$ |
| HbA$_{1c}$ (%) | $5.0 \pm 0.3$ |
| Resting metabolic rate (MJ/day) | $7.2 \pm 0.7$ |

Values represent means $\pm$ SD, $n = 12$. BMI, body mass index; HbA$_{1c}$, glycated haemoglobin.

the study, subjects visited the University for two pre-testing visits. During pretest 1, subjects arrived after an overnight fast and an oral glucose tolerance test (OGTT) was performed followed by a one-repetition unilateral maximum (1RM) strength test of both legs on the leg press and leg extension machine (Technogym, Rotterdam, the Netherlands). During pretest visit 2, subjects arrived after an overnight fast after which they were first asked to empty their bladder and bowels. Thereafter, for each subject dual-energy X-ray absorptiometry (DXA), whole-body magnetic resonance imaging (MRI) and single slice computed tomography (CT) scans were performed. Subsequently, a fasted blood and saliva sample were obtained before $D_2O$ loading was initiated and continued for the remainder of the day. The subjects received a standardized diet for the entire day and stayed at the University overnight. The morning after (bedrest day 1), a fasted blood and saliva sample were collected and a muscle biopsy was taken from both legs. Thereafter, subjects were provided with breakfast at ∼09.00 h to start the 14 day bed rest period. During the 14 day bed rest period, food was provided regularly (breakfast, lunch, dinner and snacks in between) and during three times a day (∼11.00, ∼15.00 and ∼20.00 h) unilateral blood flow restriction (BFR) was applied. Blood and saliva samples were collected during the bed rest period and every day a bolus of $D_2O$ was provided. On bedrest day 15, a fasted blood and saliva sample were collected and a muscle biopsy was taken from both legs. Thereafter, participants were transported in a wheelchair to undergo a DXA, whole-body MRI and CT scan after which they were allowed to stand up and walk again. The day after (post test), participants arrived again after an overnight fast in the laboratory to perform an OGTT and repeat the 1RM unilateral strength tests of both legs on the leg press and leg extension machine.

### Screening

During the screening visit, subjects arrived at the laboratory after an overnight fast and having refrained

**Table 2. Study timeline and procedures**

| Visits | Procedures |
|---|---|
| Screening | Eligibility criteria, informed consent, medical questionnaire, anthropometrics, blood measurements, basal metabolic rate, 1RM estimations |
| Pretest 1 | Body mass assessment, OGTT assessment, 1RM measurement |
| Pretest 2 (pre-bed rest day) | Body mass assessment, urine collection, DXA, MRI and CT scanning, $D_2O$ loading, standardized food intake |
| *14 days of bed rest* | *Every day*: standardized food intake, one-legged blood flow restriction, saliva and urine collection, $D_2O$ intake |
| Day 1 | Blood and muscle biopsy sampling |
| Days 3, 5, 7, 9, 11, 13 | Blood sampling |
| Day 15 | Blood and muscle biopsy sampling, DXA, MRI and CT scanning |
| Post test | Body mass assessment, OGTT assessment, 1RM measurement |
| Return visit (~6 weeks after the post test) | 1RM measurement |

Abbreviations: 1RM, one repetition maximum; CT, computed tomography; $D_2O$, deuterium oxide; DXA, dual-energy X-ray absorptiometry; MRI, magnetic resonance imaging; OGTT, oral glucose tolerance testing.

from alcohol and intense physical activities 48 h prior to the visit. Subjects were first fully informed of the nature and possible risks of the experimental procedures before their written informed consent was obtained. Subsequently, subjects filled out a general health questionnaire and completed a routine medical screening to ensure their eligibility to take part in the study. Body mass and height were measured and a fasting blood sample was taken to assess $HbA_{1c}$, fasting glucose concentration and D-dimer (D-dimer was also assessed every other day during bed rest to assess potential blood clot formation, but values remained low (<200 ng/mL) in all participants). Resting energy expenditure was measured by indirect calorimetry using an open-circuit ventilated hood system (Omnical, Maastricht University, Maastricht, the Netherlands), and a 1RM strength tests of both legs on the leg press and leg extension machine were performed.

**Dietary intake and physical activity prior to the bed rest period**

All subjects were asked to fill in their food intake and a physical activity questionnaire for 1 week prior to the start of the bed rest period to assess compliance to maintain a normal nutritional intake and perform no vigorous physical activities. In addition, step count was also assessed for 1 week prior to the start of the bed rest period using an accelerometer (Actical; Philips Respironics, Eindhoven, the Netherlands). The accelerometer was positioned on the right hip at the mid-clavicular line. Step count averaged $7525 \pm 1534$ steps per day during 7 days prior to bed rest. All volunteers refrained from alcohol and any sort of exhaustive physical labour and/or exercise 3 days prior to subsequent test days.

**Pretest 1 and post test**

During pretest 1 (~3 days after the screening visit) and the post test (1 day after 14 day bed rest period) subjects' body mass was measured and subsequently glucose tolerance was assessed with an OGTT. While subjects were in an overnight fasted state, an antecubital vein was cannulated to allow repeated blood sampling. Before ($t = 0$) and at $t = 10, 20, 30, 60, 90$ and 120 min following ingestion of 75 g of glucose dissolved in 200 mL solution (LemonGluc 75; Novolab, Geraardsbergen, Belgium), a blood sample was collected in a supine position to assess plasma glucose and insulin concentrations. After the OGTT, 1RM leg strength was determined for both legs separately on the leg press and extension machine as described previously (Fuchs, Kouw, et al., 2020). Finally, maximal hand grip strength was determined using a JAMAR handheld dynamometer (model BK-7498; Fred Sammons, Inc., Burr Ridge, IL, USA). Three consecutive measurements were recorded for both hands, and all three measurements were averaged to calculate mean maximal grip strength for each hand.

**Pretest 2**

On the day prior to the 14 day bed rest period, subjects reported at the laboratory after an overnight fast. Upon arrival, participants were asked to go to the toilet to make sure they emptied their bladder and bowels. Immediately thereafter, body mass and composition were measured by DXA (Hologic, Discovery A; QDR Series, Marlborough, MA, USA). The system's software package Apex version 4.0.2 was used to determine whole-body and segmental lean mass, fat mass, and bone mineral content. Muscle volume and fat infiltration were measured

by MRI at Scannexus (Brightlands Maastricht Health Campus, Maastricht, the Netherlands). Subjects were scanned in the supine position (entering the magnet head-first), with knees and hips fully extended, on a 3T MAGNETOM Prisma Fit scanner (Siemens Healthineers, Erlangen, Germany) using a whole-body coil (Siemens Healthineers). A 6 min dual-echo (T1-weighted) Dixon Vibe protocol was applied, providing a water and fat separated volumetric data set covering head to below the knees. In total eight slabs were acquired containing all individual images. Only slabs covering thigh muscles were used for analysis of thigh muscle volume and fat infiltration. Common scanning parameters were: flip angle ($\alpha$) = 10°, repetition time (TR) = 3.89 ms, echo time (TE) = 1.22/2.45 ms, bandwidth = 930 Hz/Px and 256 $\times$ 192 matrix. There was no interslice gap (0 mm). The slabs covering the thighs consisted of 88 slices with a voxel size of 2.0 $\times$ 2.0 $\times$ 3.0 mm$^3$ and were acquired during free-breathing sequences. Muscle volume was analysed for the entire thigh (including some pelvic) muscle groups combined (i.e. adductor muscle group (adductor magnus, adductor longus, adductor brevis, gracilis, pectineus and obturator externus), iliopsoas, gluteus muscle group (gluteus maximus, gluteus medius and gluteus minimus), hamstring muscle group (biceps femoris short head, biceps femoris long head, semitendinosus and semimembranosus), quadriceps femoris muscle group (rectus femoris, vastus lateralis, vastus medialis and vastus intermedius), sartorius and tensor fasciae latae). The quadriceps femoris muscle group was also analysed separately. Muscle volume analysis was performed manually using ITK-SNAP software (Yushkevich et al., 2006), as reported previously (Fuchs et al., 2023). Muscle fat infiltration was analysed as previously described (West et al., 2018).

Anatomical muscle cross-sectional area (CSA) of the whole thigh and quadriceps femoris muscle group separately were assessed via a single slice CT scan (Siemens Definition Flash; Siemens Healthineers). While subjects were lying supine, with their legs extended and their feet secured, a 2 mm thick axial image was taken 15 cm proximal to the top of the patella. The precise scanning position was marked with semi-permanent ink for replication after the 14 day bed rest period. The following scanning characteristics were used: 120 kV, 300 mA, rotation time of 1 s and a field of view of 500 mm. Tissue with Hounsfield units between −29 and 150 HU was selected as muscle tissue. CT scans were analysed by manual tracing using ImageJ software (version 2.0.0; National Institutes of Health, Bethesda, MD, USA).

After all scanning procedures, a fasted blood and saliva sample was collected followed by the ingestion of 400 mL of 70% deuterium oxide (D$_2$O; Cambridge Isotopes Laboratories, Andover, MA, USA). Subjects remained in the laboratory for the entire day, where in total eight

doses (50 mL each) of deuterium oxide were provided and ingested. For the remainder of the experiment (14 days of bed rest), participants ingested one dose each day in the evening at 20.00 h. No subjects reported any side effects of deuterated water dosing. Standardized food was provided after the first deuterium oxide dose and controlled for the remainder of the test day. Energy requirements were estimated based on indirect calorimetry data, multiplied by an activity factor of 1.4. Macronutrient composition of the diet was identical as during the bed rest period (see section 'Dietary intake during the 14 day bed rest period' for details). Finally, 24 h urine was collected for assessment of nitrogen balance (see section 'Blood, saliva, urine and muscle sampling' for details). Subjects stayed overnight at the University after which the 14 day bed rest period started.

### Two weeks of bed rest

After an overnight fast, a fasted blood and saliva sample were collected and a muscle biopsy was taken from both legs (between 07.00 and 08.00 h). Thereafter, subjects were provided with breakfast at ~09.00 h to start the 14 day bed rest period. During the entire 14 day bed rest period, standardized food was provided every day at breakfast, lunch and dinner time and subjects were allowed to consume some provided snacks at any time throughout the day. Blood flow restriction was applied daily three times a day (at ~11.00, ~15.00 and ~20.00 h) with a lower extremity pressure cuff (SC10; Hokanson, Bellevue, WA, USA) that was placed around the most proximal portion of each thigh while the subject was lying on bed. One leg was repeatedly (3 $\times$ 5 min) occluded (200 mmHg) during each session, whilst the other leg served as a control (sham: 10 mmHg).

Blood, saliva and urine samples were collected during the bed rest period and every day a bolus of D$_2$O was provided. Subjects were not allowed to leave the bed for any reason and all hygiene and sanitary activities were performed lying in bed. Subjects were permitted to use a pillow and elevation of the bed-back to perform their daily activities. Subjects were monitored continuously by the research team and compliance to the bed rest protocol was controlled using video monitoring; however, subjects themselves were never visible on camera (only the other parts in the room) and only in the case of irregularities were videos checked to ensure subjects' privacy. Every morning, subjects were woken at ~08.00 h and lights were turned off at ~23.30 h. On bedrest day 15, subjects were woken, after which they were first asked to fully empty their bladder and bowels using a plastic urinal and a bed pan (whilst still in bed). Subsequently, a fasted blood and saliva sample were collected and a muscle biopsy was taken from both legs. Thereafter, participants were transported (via a wheel-chair) to undergo a second DXA,

whole-body MRI and CT scan (as described in section 'Pretest 2') after which they were allowed to stand up and walk.

### Dietary intake during the 14 day bed rest period

During the entire bed rest period, dietary intake was strictly controlled. Meals were prepared and cooked at the University and served three times daily (breakfast, lunch, dinner) at standardized time points. Energy requirements were estimated based on indirect calorimetry data, multiplied by an activity factor of 1.3. Average daily energy intake over 14 days was 9428 $\pm$ 962 kJ/day (2246 $\pm$ 230 kcal/day). Protein intake was 1.0 $\pm$ 0.0 g/kg/day (80 $\pm$ 12 g/day), contributing to 15 $\pm$ 1% of total daily energy intake. Average fat intake was 75 $\pm$ 11 g/day, contributing to 31 $\pm$ 2% of total daily energy intake. Average carbohydrate intake was 296 $\pm$ 26 g/day, contributing to 55 $\pm$ 4% of total daily energy intake. The subjects consumed all of their meals at the scheduled times.

### Blood flow restriction protocol

The blood flow restriction protocol, applied three times daily (in the morning, afternoon and evening), was performed while the subjects were in a supine position on the bed. Blood flow restriction was applied with a pressure cuff (SC10; Hokanson) that was placed on the most proximal part of each thigh, and connected to a rapid cuff inflator (ID, Maastricht University, The Netherlands). Blood flow restriction, for one leg (the BFR leg), was initiated by inflating the cuff to a pressure of 120 mmHg for 30 s, followed by 10 s of deflation. This procedure was then repeated three more times in total while cuff pressure was increased with 20 mmHg increments (140, 160 and 180 mmHg), before finally reaching the target pressure of 200 mmHg which was maintained for 5 min after which the pressure was released again (a protocol/external compressive force that has been applied previously: Fry et al., 2010; Fujita et al., 2007; Gundermann et al., 2012; Kubota et al., 2008; Nyakayiru et al., 2019). This protocol was repeated in total three times during each session (nine times in total per day) with 1.5 min of rest in between. During the 1.5 min rest period, the pressure cuff was increased to 10 mmHg (sham) for the other leg (CON). The CON leg served as the within-subject control by not receiving a meaningful blood flow restriction stimulus. Subjects were led to believe that the aim of the study was to compare a strong with a weak blood flow restriction protocol. The leg that received the daily BFR treatment was randomized based on the dominant leg and kept the same during the entire 14 day bed rest period. To quantify the level of ischaemia/hypoxia in the BFR leg during blood flow restriction, oxygen saturation was measured by a pulse oximeter placed on the large toe of each participant. Oxygen saturation was within the normal range (98–100%) at baseline prior to cuff inflation in all subjects, but reduced to an unmeasurable range when the final pressure of 200 mmHg was reached. Oxygen saturation returned to baseline values in the 30–60 s after pressure was released from the cuff.

### Return visit

Subjects were asked to return to the laboratory ∼6 weeks after the post test to assess their 1RM on the leg press and leg extension as well as hand grip strength.

### Blood, saliva, urine and muscle sampling

During all pretests, the post test and every other day (days 1, 3, 5, 7, 9, 11, 13 and 15) during the 14 day bed rest period, blood samples (10 mL) were collected in EDTA containing tubes and centrifuged at 1000 $g$ and 4°C for 10 min. Aliquots of plasma were frozen in liquid nitrogen and stored at −80°C until analysis. During pretest 2 and every day during the 14 day bed rest period, saliva (in the fasted state in the morning) and urine samples were collected. To collect saliva, subjects lightly chewed on a cotton swab (Celluron, Hartmann, Germany) for sufficient time to saturate the cotton swab with saliva. The swab was then removed and depressed using a syringe to extract the saliva into a sample tube. After collection, saliva was frozen in liquid nitrogen and stored at −80°C until analysis. For urine collection, 24 h urine was collected starting from the second voiding of the day until the first voiding on the day after. Urine was collected into 2 L containers with 10 mL of 4 M hydrochloric acid (HCl) to prevent nitrogen loss through evaporation. After assessment of total daily urine volume, urine was gently mixed and aliquots of urine were snap-frozen in liquid nitrogen and subsequently stored at −80°C. Finally, before and after bed rest, a muscle biopsy was collected from the M. vastus lateralis of both legs with a Bergström needle under local anaesthesia (Bergstrom, 1975). Any visible non-muscle tissue was directly removed, and part of the biopsy sample was embedded in Tissue-Tek (4583; Sakura Finetek, Zoeterwoude, the Netherlands) before being frozen in liquid nitrogen-cooled isopentane. All remaining muscle tissue was immediately frozen in liquid nitrogen. Muscle samples were subsequently stored at −80°C until subsequent analysis.

### Analysis of plasma, body water enrichment and nitrogen balance

Plasma glucose and insulin concentrations, and HbA$_{1c}$ were analysed as previously described (Fuchs et al., 2019).

Body water enrichment was analysed using the saliva samples collected during pretest 2 and throughout the 2 weeks of bed rest, as described previously (Fuchs, Kouw, et al., 2020). To assess nitrogen balance, the Dumas combustion method was used to determine nitrogen content in the urine using the vario MAX cube CN (Elementar Analysensysteme, Langenselbold, Germany). Total daily nitrogen excretion was calculated from total urinary nitrogen excretion plus an estimated 10% to account for normal losses via faeces and other miscellaneous losses (Drummen et al., 2020). Nitrogen balance was calculated as the difference between nitrogen intake [protein intake (g)/6.25] and total nitrogen excretion and was used to determine daily nitrogen balance.

## Skeletal muscle analyses

**Muscle protein synthesis.** For measurement of $[^2H]$ alanine enrichment in the intramuscular mixed muscle protein pool, ∼55 mg wet muscle was freeze dried. Collagen, blood and other non-muscle fibre material were removed from the muscle fibres under a light microscope. The isolated muscle fibre mass (∼10 mg) was weighed, and 35 volumes (35 times dry weight of isolated muscle fibres wet:dry ratio) of ice-cold 2% perchloric acid was added. The tissue was then sonicated and centrifuged. The protein pellet was collected and washed with three additional 1.0 mL washes of 2% perchloric acid, dried and hydrolysed in 6 M HCl at 110°C for 15–18 h. The hydrolysed protein fraction was dried under a nitrogen stream while being heated at 110°C. After being dissolved with a 50% acetic acid solution, samples were passed over Dowex exchange resin (AG 50W-X8, 100–200 mesh hydrogen form; Bio-Rad, Hercules, CA, USA), washed with 0.1 M HCl and eluted by using 2 M $NH_4OH$. Thereafter, the eluate was dried, and the purified amino acids were derivatized to their N(O,S)-ethoxycarbonyl ethyl esters. The derivatized samples were measured using a gas chromatography-isotope ratio mass spectrometer (MAT 253; Thermo Fisher Scientific, Bremen, Germany) equipped with a pyrolysis oven using a 60 m DB-17MS column and 5 m precolumn (No. 122-4762; Agilent, Santa Clara, CA, USA) and GC-Isolink. Ion masses 2 and 3 were monitored to determine the $^2H/^1H$ ratios of muscle protein-bound alanine. A series of known standards were applied to assess linearity of the mass spectrometer and to control for the loss of tracer.

**Muscle fibre characteristics.** For immunohistochemistry, muscle biopsies were cut into 5 $\mu$m thick cross-sections using a cryostat (CM 3050, Leica Biosystems, Nussloch, Germany) and stored at −80°C for subsequent analyses. Samples from both pre and post bed rest of both the BFR and CON leg were mounted on the same glass slide. Immunohistochemistry was performed to assess type I and type II muscle fibre cross-section, as described previously (Nilwik et al., 2013). In short, muscle samples were air dried for 30 min followed by a 5 min fixation step in acetone (VWR Chemicals, Vienna, Austria), and a 30 min blocking step in 3% BSA in 0.1% Tween/PBS. Between incubations, slides were washed once for 5 min in 0.1% Tween/PBS and twice for 5 min in PBS. The following antibodies were dissolved in a 0.1% BSA/0.1% Tween/PBS staining solution. Slides were incubated for 30 min with anti-myosin heavy chain type 1 (A4.840, DSHB, 1:25) and anti-Laminin (L9393, Sigma-Aldrich, Darmstadt, Germany, 1:50). Next, samples were incubated with appropriate secondary antibodies, GAMIgM Alexa 555 (A21426, Invitrogen, Carlsbad, CA, USA, 1:500) and GARIgG Alexa 647 (A21238, Invitrogen, 1:400). Finally, slides were mounted with Mowiol (Calbiochem, Amsterdam, the Netherlands) and covered by a glass cover slip. All sections were recorded using a CorrSight fluorescence microscope using a Zeiss 20× NA0.8 AIR objective, an Andromeda spinning disc module and Hamamatsu ORCA-Flash4.0 V2 camera, resulting in a pixel size of 326 nm (FEI Company, Eindhoven, the Netherlands). Quantitative analyses were performed using ImageJ version 1.46d software package (National Institutes of Health; Strandberg et al., 2010), as described previously (Nilwik et al., 2013). For determination of muscle fibre CSA and fibre type distribution, the average number of type I and type II muscle fibres included was as follows: BRF leg: pre 125 ± 64 and 277 ± 174, post 94 ± 40 and 200 ± 117, and CON leg: pre 109 ± 35 and 250 ± 92, post 103 ± 59 and 177 ± 88, respectively.

**Western blotting and mRNA analyses.** For western blot analysis, ∼40 mg of muscle sample was used and was analysed as described previously (Fuchs, Smeets, et al., 2020). For mRNA analysis, ∼20 mg of muscle sample was used and was analysed as described previously (Fuchs, Kouw, et al., 2020).

**Skeletal muscle oxidative capacity.** For mitochondrial enzyme activities, ∼50 mg of muscle was weighed. To assess citrate synthase activity, muscle sample was homogenized in SET buffer (2.5%) using potter tubes and pestles. Subsequently, the homogenate was frozen and thawed four times and, subsequently, sonicated to break mitochondrial membranes. Citrate synthase activity was measured by assessing the condensation reaction of acetyl-CoA and oxaloacetate to form citrate and CoA-SH. CoA-SH reacts with 5.5′-dithiobis (2-nitrobenzoic acid) (DTNB) to CoA-DTNB, which was measured photometrically at 412 nm in a COBAS-FARA semiautomatic analyser (COBAS-FARA; Roche, Basel, Switzerland).

For assessment of cytochrome C oxidase activity, muscle samples were homogenized in SET buffer (5.0%) using potter tubes and pestles. Cytochrome C oxidase activity was measured by assessing the oxidation of reduced cytochrome C photometrically at 550 nm in a COBAS-FARA semi-automatic analyser (COBAS-FARA; Roche). Both enzyme activities are expressed as micromoles of product (citrate and oxidized cytochrome C, respectively) generated per gram of wet muscle tissue per min ($\mu$mol/g/min) (Gohil et al., 1981; Shepherd & Garland, 1969).

## Calculations

Total area under the curve (AUC) for plasma glucose and insulin concentrations from the OGTT before and after the 14 day bed rest period was calculated using the trapezoidal method. In addition, total incremental AUC (iAUC) was also calculated for OGTT values. The homeostatic model assessment of insulin resistance (HOMA-IR) was calculated based on fasted glucose and insulin concentrations before (pretest 2) and at the end of bed rest (day 15).

Mixed muscle protein fractional synthetic rate (FSR) was determined using the incorporation of [²H]alanine into muscle proteins and mean body water deuterium enrichment corrected by a factor of 3.7 based on the deuterium labelling during *de novo* alanine synthesis (Holwerda et al., 2018). FSR was calculated (for both the CON and BFR leg) using the standard precursor-product method:

$$\text{FSR}\ \left(\% \cdot \text{d}^{-1}\right) = \left(\frac{E_{m2} - E_{m1}}{E_{\text{precursor}} \times t}\right) \times\ 100\%$$

where $E_{m1}$ and $E_{m2}$ are the mixed muscle protein-bound enrichments on days 1 and 15, respectively, in each leg; $E_{\text{precursor}}$ represents mean body water deuterium enrichment corrected by a factor of 3.7; and $t$ represents the time between biopsies on day 1 and 15.

Absolute synthetic rate of mixed muscle protein (ASR; g/d) for quadriceps muscle in both the CON and BFR leg was calculated as:

$$\text{ASR}\ \ \left(\text{g} \cdot \text{d}^{-1}\right) = \left(\frac{\text{FSR}}{100}\right) \times \text{quadriceps muscle mass}$$
$$\times 1000\ \times \left(\frac{17.2}{100}\right)$$

with average quadriceps muscle mass over the 2 week bed rest period in kilograms (which can be calculated given that skeletal muscle density is ∼1.04 g/cm) and where a total protein content of 17.2% is assumed (Snyder et al., 1975).

Absolute protein breakdown rate (ABR) was estimated as follows:

$$\text{ABR}\ \left(\text{g} \cdot \text{d}^{-1}\right) = \left(\frac{\text{FBR}}{100}\right) \times \text{quadriceps muscle mass}$$
$$\times 1000\ \times \left(\frac{17.2}{100}\right)$$

where fractional breakdown rate (FBR) is calculated as FBR = FSR − fractional growth rate (FGR), with the FGR assumed to be percentage quadriceps muscle mass loss per day over 2 weeks. Finally, net protein balance was calculated as follows: ASR − ABR.

## Statistical analysis

All data in the text and tables are expressed as means ± standard deviation (SD). All data in graphs are expressed as means ± SD, 95% confidence interval (CI) and/or individual values. Changes in saliva (body water ²H), plasma glucose and insulin concentrations during bed rest were analysed using one-way repeated-measures ANOVA with time as the within-subjects factor. A two-factor (with treatment and time as within-subject factors) repeated-measures ANOVA was performed for the analysis of leg lean mass (DXA), muscle volume (MRI), muscle CSA (CT), muscle strength data, muscle fat infiltration, anabolic signalling, gene expression and markers of oxidative capacity. Plasma OGTT data over time were analysed with repeated-measures ANOVA with time and intervention (i.e. pre *vs.* post bed rest) as within-subject factors. Skeletal muscle fibre type CSA and distribution data were analysed with repeated-measures ANOVA with treatment, time and type as within-subject factors. Finally, a Student's paired *t* test (two-sided) was performed for the analysis of values before and after bed rest [i.e. all other (apart from legs) body composition data, AUC and iAUC data, and HOMA-IR] as well as differences between the CON and BFR leg during the 2 week bed rest period [i.e. mixed muscle protein synthesis (FSR), ASR, FBR, ABR and net protein balance]. In case of a significant main effect of time or interaction, Bonferroni corrected pairwise comparisons were performed where appropriate. Statistical significance was set at $P < 0.05$. All calculations were performed using SPSS (version 26.0, IBM Corp., Armonk, NY, USA).

## Results

### Body composition

Bed rest resulted in 1.8 ± 1.0 kg (representing 3.0 ± 1.5%) lean body mass loss ($P < 0.001$; Table 3). In addition, a decrease in total fat mass was observed. Whole-body

**Table 3. Body composition before (Pre) and after (Post) 14 days of strict bed rest in healthy, young men**

|  | Pre | Post | Difference | *P* value |
|---|---|---|---|---|
| Total body mass (kg) | 80.4 ± 12.5 | 78.2 ± 12.2 | −2.1 ± 1.1 | <0.001 |
| Total lean body mass (kg) | 59.4 ± 8.1 | 57.6 ± 7.9 | −1.8 ± 1.0 | <0.001 |
| Lean mass trunk (kg) | 28.4 ± 3.9 | 27.9 ± 4.0 | −0.5 ± 0.9 | 0.069 |
| CON leg lean mass (kg) | 10.2 ± 1.6 | 9.7 ± 1.6 | −0.5 ± 0.2 | <0.001 |
| BFR leg lean mass (kg) | 10.2 ± 1.7 | 9.6 ± 1.6 | −0.6 ± 0.2 | <0.001 |
| L arm lean mass (kg) | 3.6 ± 0.6 | 3.5 ± 0.6 | −0.1 ± 0.1 | 0.021 |
| R arm lean mass (kg) | 3.7 ± 0.6 | 3.6 ± 0.6 | −0.1 ± 0.1 | <0.001 |
| Appendicular lean mass (kg) | 27.7 ± 4.3 | 26.4 ± 4.1 | −1.3 ± 0.4 | <0.001 |
| SMMI (kg/m$^2$) | 8.3 ± 1.0 | 7.9 ± 1.0 | −0.4 ± 0.1 | <0.001 |
| Total fat mass (kg) | 18.0 ± 5.4 | 17.7 ± 5.3 | −0.4 ± 0.4 | 0.013 |
| Whole body fat percentage (%) | 22.1 ± 4.1 | 22.3 ± 4.3 | 0.2 ± 0.5 | 0.309 |
| Bone mineral content (kg) | 2.9 ± 0.4 | 2.9 ± 0.4 | 0 ± 0 | 0.391 |

Values represent means ± SD, *n* = 12. SMMI, skeletal muscle mass index; BFR, blood flow restriction leg; CON, control leg.

fat percentage and bone mineral content did not change significantly during the bed rest period.

For both legs, bed rest resulted in a significant decline in leg lean mass of 5.3 ± 1.8% in the CON and 5.3 ± 1.7% in the BFR leg ($P < 0.001$; Table 3). In addition, a significant decline in thigh muscle volume of 5.1 ± 2.7% in the CON and 5.0 ± 2.5% in the BFR leg was observed ($P < 0.001$; Fig. 1*A*). For thigh muscle CSA, a significant decline of 6.2 ± 2.7% in the CON and 5.5 ± 2.3% in the BFR leg was shown ($P < 0.001$; Fig. 1*B*). No differences between the CON and BFR legs were observed for any of the outcome parameters ($P > 0.05$). A significant decline was observed after bed rest for quadriceps femoris muscle volume of 6.8 ± 3.7% in the CON and 6.5 ± 3.7% in the BFR leg ($P < 0.001$; Fig. 1*C*) and quadriceps femoris muscle CSA of 6.1 ± 3.9% in the CON and 5.5 ± 3.5% in the BFR leg ($P < 0.001$; Fig. 1*D*), with no differences between legs.

Skeletal muscle fibre type CSA data are shown in Fig. 2. Analysis of skeletal muscle fibre size showed a significant treatment × time × type interaction ($P = 0.032$). Further analysis revealed a significant time × type interaction for the BFR leg only ($P = 0.013$), with only a significant difference observed between type I *vs.* type II muscle fibre CSA following bed rest ($P = 0.049$). A tendency ($P = 0.062$) was observed for the decline in type I and type II muscle fibre size in the CON leg. No significant change in fibre type distribution was observed over time in both the CON as well as BFR leg following 2 weeks of strict bed rest.

A tendency ($P = 0.056$) for an increased thigh muscle fat infiltration (assessed with MRI) was observed in response to 2 weeks of bed rest. When assessing only the anterior thigh muscle groups (i.e. quadriceps femoris, sartorius and tensor fascia latae), a significant time effect was observed ($P = 0.023$), demonstrating more muscle fat infiltration in the anterior thigh muscles after bed rest

(from 4.4 ± 1.2 to 4.5 ± 1.4% in CON and from 4.0 ± 1.0 to 4.3 ± 1.1% in BFR).

## Muscle strength

In line with results for leg muscle volume, a significant decline in leg press 1RM strength was observed of 6.1 ± 8.2% in CON and of 6.7 ± 8.7% in BFR ($P = 0.024$), with no differences between treatments (time × treatment effect: $P = 0.658$; Fig. 3*A*). Similarly, 1RM leg extension strength decreased by 8.7 ± 12.4% in CON and by 10.0 ± 13.2% in BFR (time effect: $P = 0.014$), with no differences between treatments (time × treatment effect: $P = 0.594$; Fig. 3*B*). After ∼6 weeks, 1RM leg press and extension strength were not different from 1RM values obtained prior to bed rest in both the CON and BFR leg [leg press: 153 ± 37 kg in CON, 147 ± 43 kg in BFR ($P = 0.700$); leg extension: 62 ± 14 kg in CON, 61 ± 13 kg in BFR ($P = 0.105$)]. Following bed rest, no changes in handgrip strength were observed. Grip strength in the left arm averaged 41 ± 4 kg before bed rest and 41 ± 5 kg after bed rest ($P = 0.254$). Grip strength in the right arm averaged 45 ± 5 kg before bed rest and 44 ± 5 kg after bed rest ($P = 0.401$).

## Muscle protein synthesis and (extrapolated) breakdown

Analysis of daily saliva samples revealed a gradual increase in body water enrichment over time (from 0.60 ± 0.07 to 0.95 ± 0.15%; $P < 0.001$). Mixed muscle protein synthesis rates (Fig. 4) during bed rest did not differ between the BFR and CON leg (1.11 ± 0.12 *vs.* 1.08 ± 0.13%/day, respectively; $P = 0.302$). Furthermore, no differences were observed in mixed muscle protein synthesis rates between the dominant or non-dominant leg (1.09 ± 0.12 *vs.* 1.10 ± 0.14%/day, respectively; $P = 0.767$). In

addition, ASR was not significantly different between the quadriceps muscle of the BFR and CON leg ($4.7 \pm 0.8$ *vs.* $4.6 \pm 0.9$ g/day, respectively; $P = 0.496$). The fractional growth rate (i.e. percentage quadriceps muscle mass loss per day over 2 weeks) was $-0.5 \pm 0.3$ *vs.* $-0.5 \pm 0.3\%$/day for the BFR and CON leg, respectively ($P = 0.284$). As a result, FBR could be calculated and this averaged $1.57 \pm 0.18$ *vs.* $1.57 \pm 0.23\%$/day for the quadriceps muscle in the BFR and CON leg, respectively ($P = 0.939$). Subsequently, ABR averaged $6.7 \pm 1.5$ *vs.* $6.7 \pm 1.5$ g/day for the quadriceps muscle in the BFR and CON leg, respectively ($P = 0.861$). Finally, net protein balance of the quadriceps muscle averaged $-2.0 \pm 1.2$ g/day in the BFR and $-2.1 \pm 1.2$ g/day in the CON leg, with no differences between legs ($P = 0.283$).

## Anabolic signalling and gene expression

The phosphorylation status (ratio of phosphorylated to total protein) of key proteins involved in the initiation of muscle protein synthesis are presented in Fig. 5. A significant treatment effect was observed for muscle mTOR (Ser2448) phosphorylation status ($P = 0.030$; Fig. 5*A*), with no significant time or time × treatment interaction observed. No significant differences were observed for muscle p70S6K (Thr389) (Fig. 5*B*), p70S6K (Thr421/Ser424) (Fig. 5*C*), rpS6 (Ser235/236) (Fig. 5*E*) or 4E-BP1 (Thr37/46) (Fig. 5*F*) phosphorylation status. A significant time effect was observed for muscle rpS6 (Ser240/244) phosphorylation status ($P = 0.014$; Fig. 5*D*), with no significant treatment or time × treatment interaction.

Skeletal muscle mRNA expression for selected genes implicated in the regulation of muscle mass, inflammation, intracellular amino acid and glucose transport are displayed in Fig. 6. No significant time effects, treatment effects and time × treatment interactions were observed for muscle FOXO1 (Fig. 6*A*), MuRF1 (Fig. 6*B*), MAFbx (Fig. 6*C*), P70S6K (Fig. 6*E*), TNF-$\alpha$ (Fig. 6*F*), LAT1/SLC (Fig. 6*H*), SNAT2 (Fig. 6*J*;

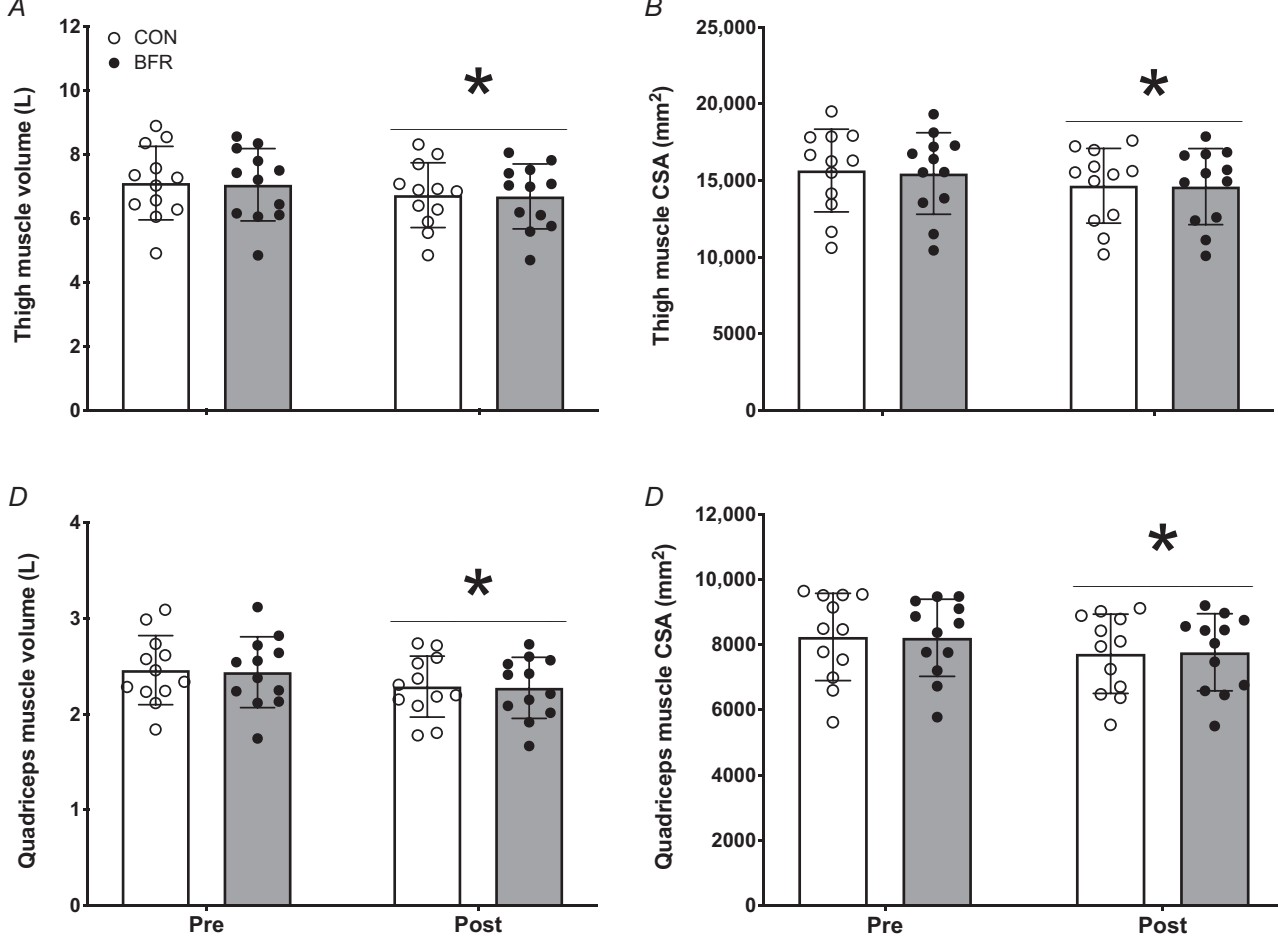

**Figure 1. Thigh muscle volume (*A*), thigh muscle CSA (*B*), quadriceps muscle volume (*C*) and quadriceps muscle CSA (*D*) before (Pre) and after (Post) 14 days of strict bed rest in healthy, young men (*n* = 12)**
Bars are means ± SD, and dots represent individual values. *Significantly different ($P < 0.001$) from pre. BFR, blood flow restriction leg; CON, control leg.

however, a tendency for a significant time effect was observed here: $P = 0.063$) or CD98 (Fig. 6*K*) mRNA expression. A significant treatment effect, but no time effect and/or time × treatment interaction, was observed for muscle mTOR (Fig. 6*D*; $P = 0.004$) and PAT1 (Fig. 6*I*; $P = 0.036$) mRNA expression. Both a significant treatment ($P = 0.040$) and time effect ($P = 0.002$), but no time × treatment interaction, was observed for muscle GLUT4 mRNA expression (Fig. 6*L*). A significant time × treatment interaction ($P = 0.026$), but no treatment or time effect, was observed for muscle IL-6 mRNA expression (Fig. 6*G*). Muscle IL-6 mRNA expression was significantly higher in the CON leg after bed rest ($P = 0.020$), with no significant differences in the BFR leg ($P = 0.660$).

## Oxidative capacity

A significant time effect was observed for both citrate synthase activity ($P = 0.018$; Fig. 7*A*) and cytochrome C oxidase activity ($P = 0.037$; Fig. 7*B*), with no significant treatment or time × treatment interactions observed. Citrate synthase activity decreased by ∼15%

(from $11.8 \pm 4.1$ to $10.0 \pm 3.2$ $\mu$mol/min/g wet weight) in the CON leg and by ∼16% (from $12.6 \pm 3.1$ to $10.6 \pm 2.8$ $\mu$mol/min/g wet weight) in the BFR leg. Cytochrome C oxidase activity decreased by ∼8% (from $41.2 \pm 18.2$ to $38.1 \pm 15.4$ $\mu$mol/min/g wet weight) in the CON leg and by ∼21% (from $46.8 \pm 15.3$ to $37.2 \pm 12.9$ $\mu$mol/min/g wet weight) in the BFR leg.

## Nitrogen balance

Dietary nitrogen intake during bed rest, derived from calculated dietary protein intake, was on average $12.8 \pm 0.1$ g/day. Daily nitrogen loss in urine averaged $14.0 \pm 1.3$ g/day. Average daily nitrogen balance was $-1.2 \pm 1.3$ g/day.

## Glycaemic control

Basal and postprandial plasma glucose concentrations observed during the OGTT before and after the 14 day

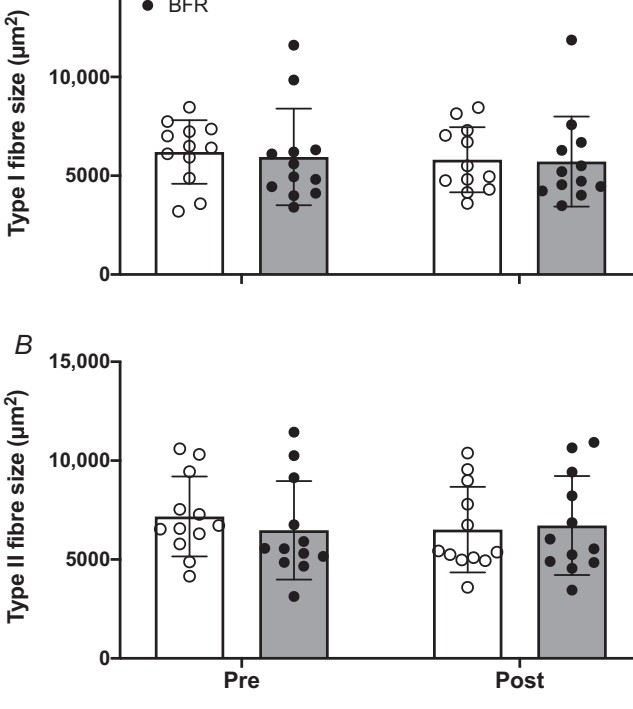

**Figure 2. Type I (*A*) and type II (*B*) muscle fibre CSA before (Pre) and after (Post) 14 days of strict bed rest in healthy, young men (*n* = 12)**
Bars are means ± SD, and dots represent individual values. BFR, blood flow restriction leg; CON, control leg.

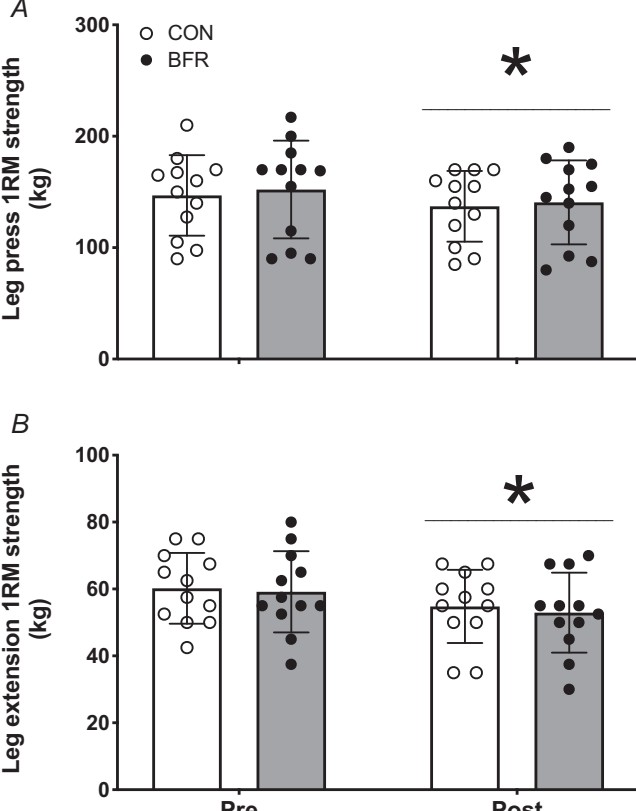

**Figure 3. Leg press (*A*) and extension (*B*) 1RM strength before (Pre) and after (Post) 14 days of strict bed rest in healthy, young men (*n* = 12)**
Bars are means ± SD, and dots represent individual values.
*Significantly different ($P < 0.05$) from pre. 1RM, one repetition maximum; BFR, blood flow restriction leg; CON, control leg.

bed rest period are displayed in Fig. 8A. A time effect was observed ($P < 0.001$) and a tendency for a significant interaction between OGTT pre and post bed rest ($P = 0.051$). For plasma glucose, AUC and iAUC (Fig. 8A inset) did not differ between the two tests ($P > 0.05$). Insulin concentrations observed during the OGTT before and after the 14 day bed rest period are displayed in Fig. 8B. A time ($P < 0.001$) and bed rest ($P = 0.047$), but no interaction, effect ($P = 0.269$) was observed between OGTT pre and post bed rest. Plasma insulin concentrations showed a tendency for a significant increase in AUC ($P = 0.053$) and a significant increase in iAUC ($P = 0.039$; Fig. 8B inset) following bed rest. Fasting plasma glucose concentrations (Fig. 9) averaged $4.7 \pm 0.3$ mmol/L prior to bed rest and did not change during the bed rest period ($P > 0.05$). For plasma insulin concentrations (Fig. 9), a significant time effect ($P = 0.018$) was observed and fasting insulin concentrations had increased from $7.8 \pm 5.0$ mU/L prior to bed rest to $10.7 \pm 4.4$ mU/L at the end of 14 days of bed rest, but further analysis did not reveal any significant differences between plasma insulin concentrations before and at the end of bed rest ($P > 0.05$). The HOMA-IR value, which indicates insulin resistance, showed an ~34% increase throughout the bed rest period ($P = 0.023$; Fig. 9 inset).

## Discussion

In the present study we observed that 2 weeks of bed rest resulted in substantial declines in whole-body insulin sensitivity, skeletal muscle mass, strength and muscle oxidative capacity. The application of daily blood flow restriction during bed rest did not increase muscle protein synthesis rates and did not attenuate the decline in skeletal

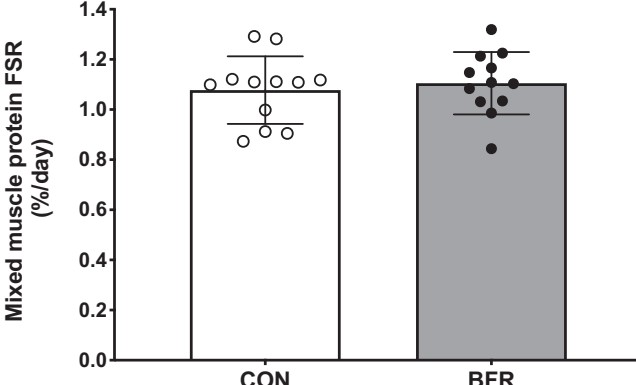

**Figure 4. Mixed muscle protein synthesis rates, expressed as fractional synthetic rate (FSR) per day, during 14 days of strict bed rest in healthy, young men (n = 12)**
Bars are means ± SD, and dots represent individual values. BFR, blood flow restriction leg; CON, control leg.

muscle quantity, muscle quality and muscle function following 2 weeks of strict bed rest.

The impacts of prolonged bed rest on metabolic health, skeletal muscle mass and strength have been well established. In our lab, we have previously observed that just a single week of bed rest induces a 1.4 kg loss of lean body mass and an ~8% decline in 1RM leg muscle strength in healthy adults (Dirks et al., 2016). Others have reported similar findings (i.e. an ~1.3–2.0 kg lean body mass loss with a concomitant ~9–16% decline in leg strength) following 7–14 days of bed rest (Arentson-Lantz, Galvan, Ellison, et al., 2019; Arentson-Lantz, Galvan, Wacher, et al., 2019; Bamman et al., 1998; Deutz et al., 2013; English et al., 2016; Ferrando et al., 2010; Kortebein et al., 2007, 2008). The present data are in line with these previous findings, with a 1.8 kg loss of lean body mass (Table 3) and a concomitant ~ 7–11% decline in leg strength (Fig. 3) observed following 14 days of bed rest. Our findings also support previous estimations that healthy adults tend to lose ~0.5% muscle mass per day during a prolonged period of disuse (Wall & van Loon, 2013).

Blood flow restriction has been postulated as an effective strategy to attenuate the loss of appendicular muscle mass and strength during a period of disuse (Loenneke et al., 2012a; Martin et al., 2022; Patterson et al., 2019). However, data on the proposed efficacy of blood flow restriction to preserve muscle mass and function remain discrepant, with some studies (Kakehi et al., 2020; Kubota et al., 2008, 2011; Takarada et al., 2000) reporting benefits of blood flow restriction whereas others have been unable to detect muscle mass or strength preservation during a period of immobilization (Slysz et al., 2020). It has been suggested that the discrepancy between studies could be due to blood flow restriction being more effective in scenarios involving more severe atrophy or wasting and/or the applied frequency or volume of the blood flow restriction treatment (Slysz et al., 2020). In the present study, we extended on these previous reports by investigating the effect of daily blood flow restriction (three sessions per day) on muscle protein synthesis rates, appendicular muscle mass, leg strength and muscle oxidative capacity following a distinct disuse model, prolonged bed rest.

We applied deuterium oxide administration to first assess muscle protein synthesis rates (Holwerda et al., 2018; Miller et al., 2020) throughout the 2 weeks of bed rest in both the blood flow restricted and control leg (Fig. 4). Daily muscle protein synthesis rates averaged $1.11 \pm 0.12$ and $1.08 \pm 0.13$%/day in the blood flow restricted and control leg, respectively, with no differences between legs. The relatively low daily muscle protein synthesis rates in both legs agree with muscle protein synthesis rates reported previously (0.8–1.3%/day) under various conditions where physical activity is minimized

(Kilroe et al., 2020; McGlory et al., 2018; Shad et al., 2019; Smeuninx et al., 2020; Weijzen et al., 2023). Previously, we have shown that under normal resting conditions (also in healthy young males), muscle protein synthesis rates average around ~1.6–1.7%/day (Holwerda et al., 2018). Assuming that muscle protein breakdown rates do not change considerably during bed rest (Brook et al., 2022; Nunes et al., 2022; Pavis et al., 2023; Phillips & McGlory,

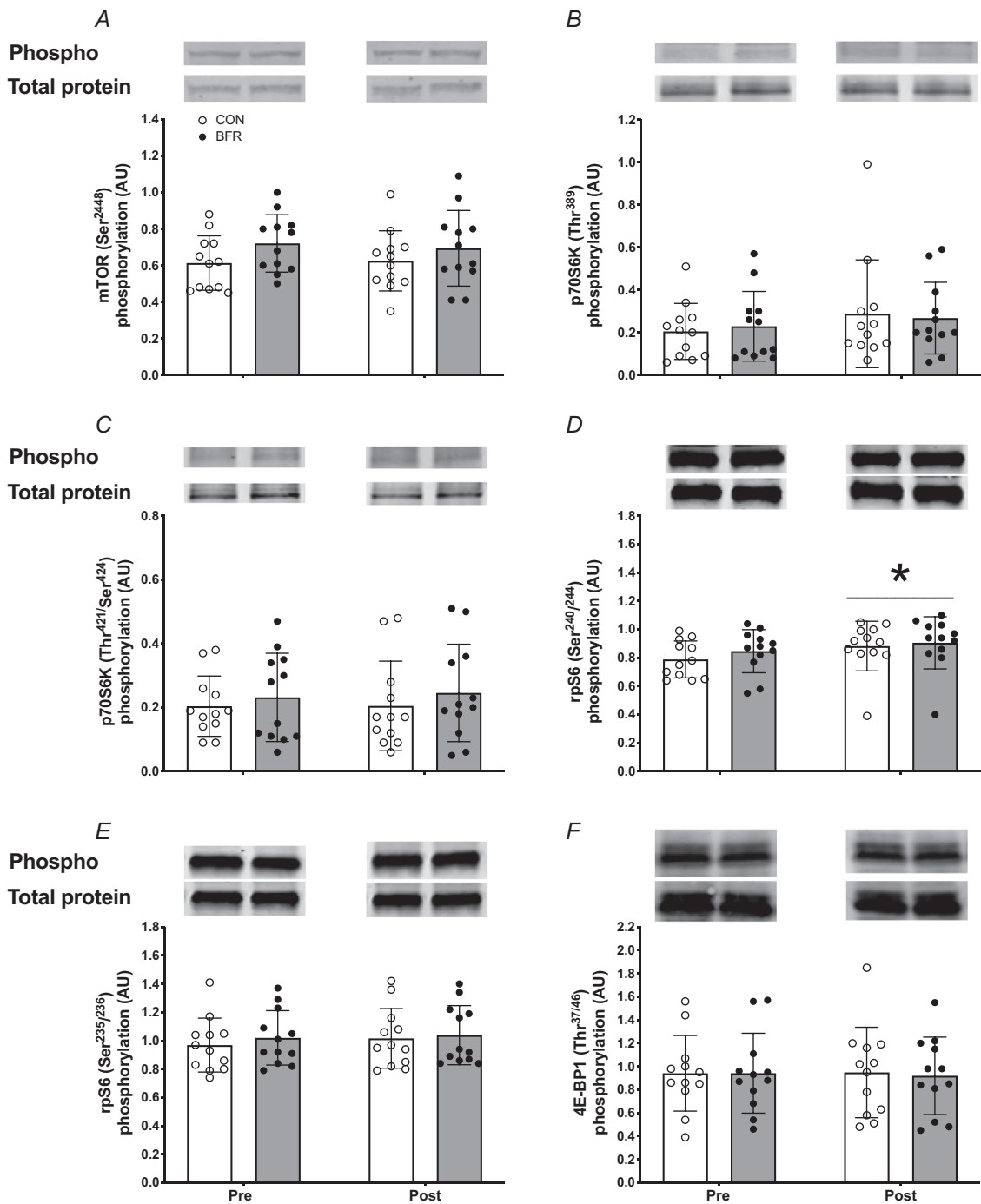

**Figure 5. Skeletal muscle phosphorylation status (ratio of phosphorylated to total protein) of mTOR (Ser2448) (*A*), p70S6K (Thr389) (*B*), p70S6K (Thr421/Ser424) (*C*), rpS6 (Ser240/244) (*D*), rpS6 (Ser235/236) (*E*) and 4E-BP1 (Thr37/46) (*F*) before (Pre) and after (Post) 14 days of strict bed rest in healthy, young men (*n* = 12)**
Bars are means ± SD, and dots represent individual values. *Significantly different (*P* = 0.014) from pre. BFR, blood flow restriction leg; CON, control leg.

2014; Wall & van Loon, 2013), this would imply that muscle protein synthesis rates had declined by at least 0.5%/day throughout the 2 weeks of bed rest. This agrees with previous studies showing an ∼30–50% decline in muscle protein synthesis rates during bed rest of a similar (10–14 days) duration (Ferrando et al., 1996; Kortebein et al., 2007). The absence of a stimulating effect of three daily sessions of blood flow restriction during prolonged bed rest seems to be in line with our inability to detect an increase in muscle protein synthesis rates following acute

recovery from a single session of blood flow restriction (Nyakayiru et al., 2019). Consequently, we must conclude that a single session or more prolonged daily application of blood flow restriction does not increase muscle protein synthesis rates.

As disuse-induced muscle atrophy has been largely attributed to the observed decline in both postabsorptive as well as postprandial muscle protein synthesis rates (Breen et al., 2013; Brook et al., 2022; English et al., 2016; Kilroe et al., 2020; McGlory et al., 2018; Shad

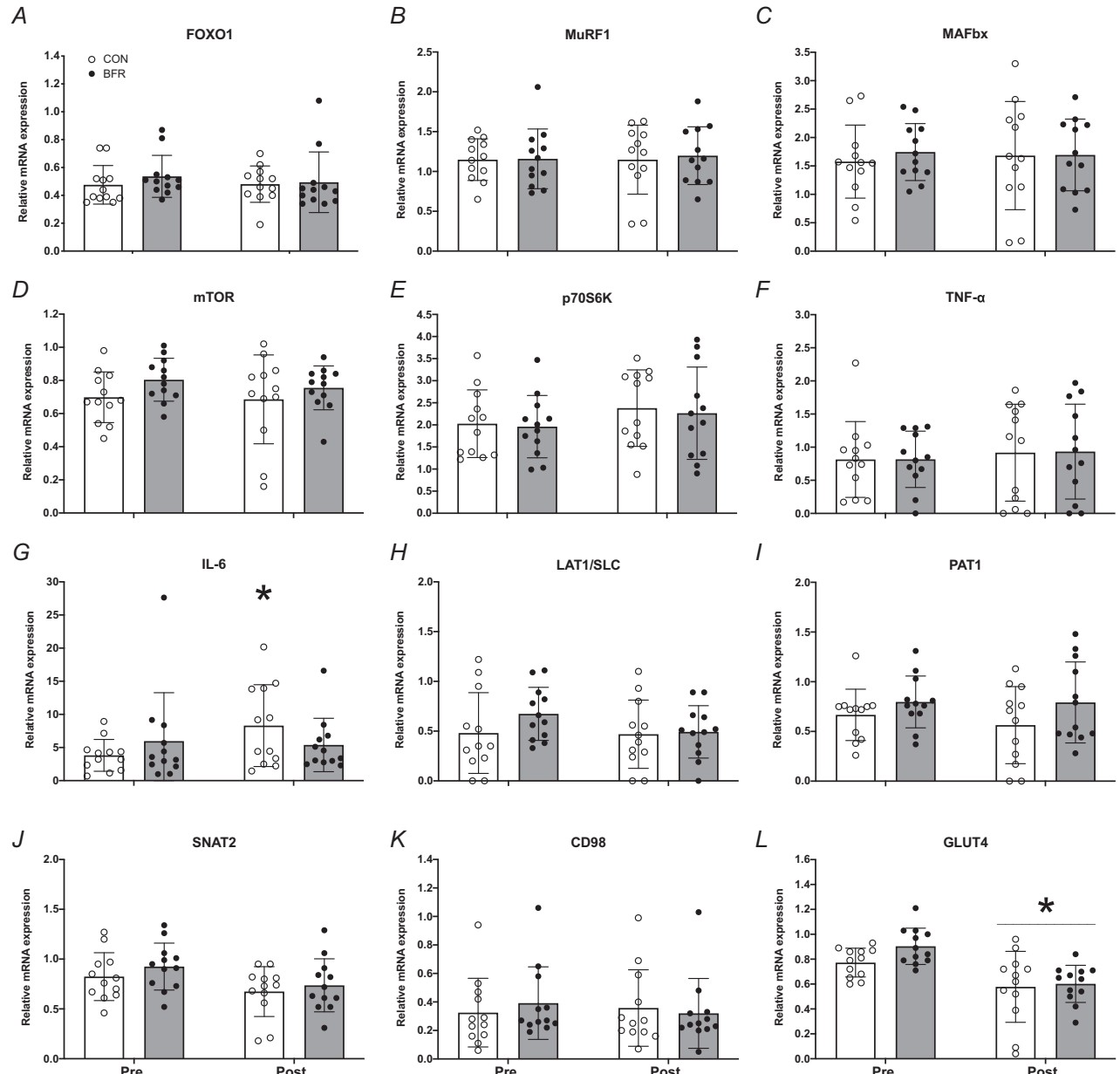

**Figure 6. Skeletal muscle mRNA expression of FOXO1 (*A*), MuRF1 (*B*), MAFbx (*C*), mTOR (*D*), p70S6K (*E*), TNF-*α* (*F*), IL-6 (*G*), LAT1/SLC (*H*), PAT1 (*I*), SNAT2 (*J*), CD98 (*K*) and GLUT4 (*L*) before (Pre) and after (Post) 14 days of strict bed rest in healthy, young men (*n* = 12)**
Bars are means ± SD, and dots represent individual values. *Significantly different ($P < 0.05$) from pre. BFR, blood flow restriction leg; CON, control leg.

et al., 2019; Smeuninx et al., 2020; Wall & van Loon, 2013; Wall et al., 2016), it may not be surprising that we did not detect an attenuated decline in muscle mass in the blood flow restricted compared to the control leg. Leg lean mass (Table 3), leg muscle volume (Fig. 1*A* and *C*) and leg muscle CSA (Fig. 1*B* and *D*) had declined substantially following bed rest, with no preservation of muscle mass following blood flow restriction on any of these parameters. The overall declines in leg muscle volume and CSA were accompanied by a considerable decline in leg press and leg extension strength, which again did not differ between the leg with blood flow restriction treatment and the control leg (Fig. 3*A* and *B*). Clearly, the application of blood flow restriction three times daily was unable to preserve muscle mass as well as strength during prolonged bedrest.

Previous work has suggested that blood flow restriction could attenuate the disuse-induced loss in muscle mass by activating anabolic and/or inhibiting proteolytic signalling cascades in skeletal muscle tissue (Kakehi et al., 2020; Nakajima et al., 2016). Alongside the absence of differences in muscle protein synthesis rates between the blood flow restricted and control leg, we did not observe any differences in baseline molecular signalling (Figs 5 and 6) following the daily application of blood flow restriction. These data align with our findings in muscle protein synthesis rates and our previous work in which we assessed the impact of a single session of blood flow restriction on anabolic signalling responses in muscle tissue (Nyakayiru et al., 2019). As muscle disuse is often accompanied by (subclinical) inflammation (Crossland et al., 2019; Drummond et al., 2013), we also assessed

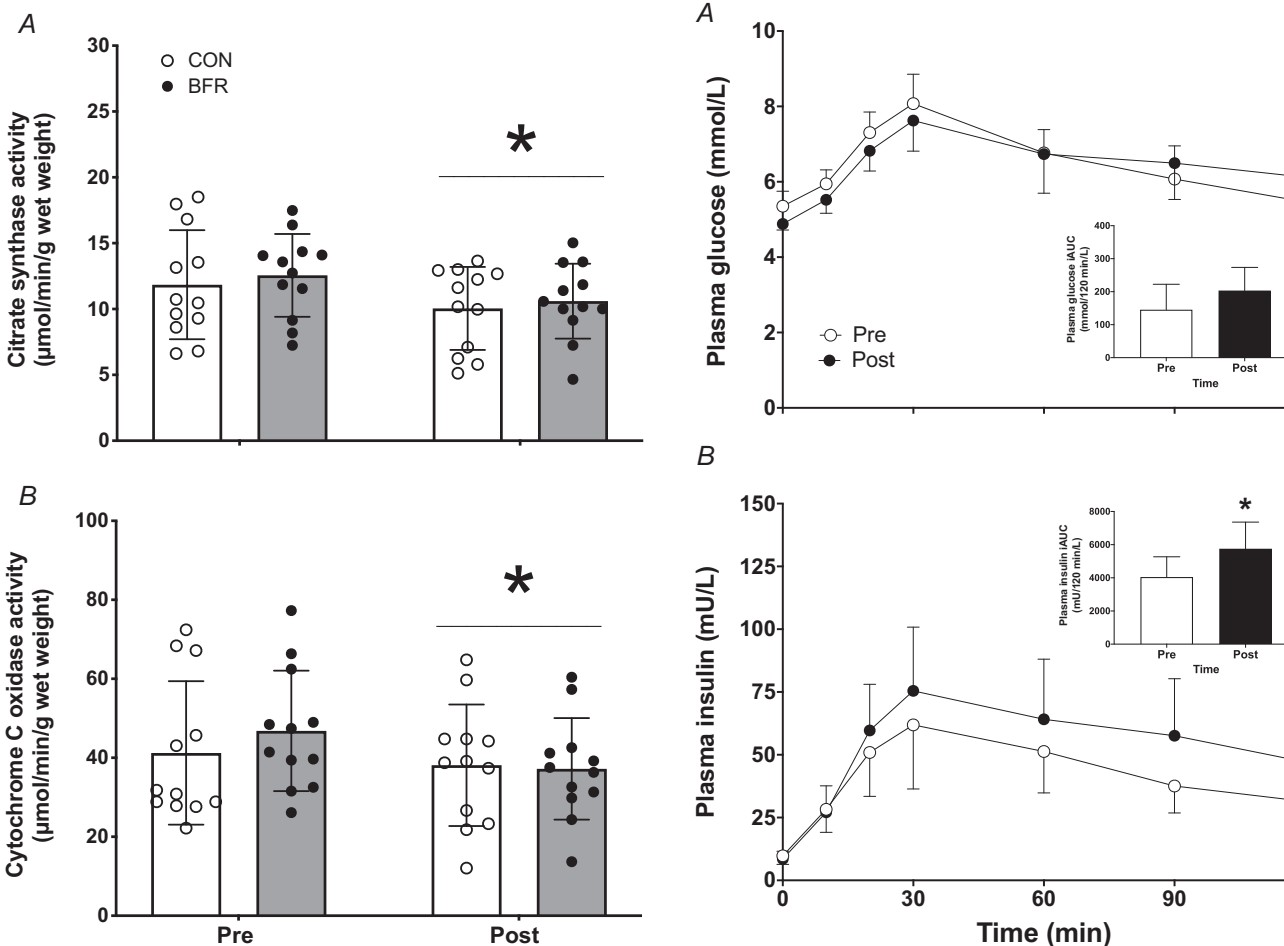

**Figure 7. Skeletal muscle citrate synthase (*A*) and cytochrome C oxidase (*B*) activity before (Pre) and after (Post) 14 days of strict bed rest in healthy, young men (*n* = 12)**
Bars are means ± SD, and dots represent individual values. *Significantly different (*P* < 0.05) from pre. BFR, blood flow restriction leg; CON, control leg.

**Figure 8. Plasma glucose (*A*) and insulin (*B*) concentrations over time, and plasma glucose iAUC (*A* inset) and insulin iAUC (*B* inset) during an OGTT before (Pre) and after (Post) 14 days of strict bed rest in healthy, young men (*n* = 12)**
Values represent means ± 95% CI. *Significantly different (*P* = 0.039) from pre.

markers of local inflammation (Fig. 6*F* and *G*). Whereas we did not observe any differences in TNF-$\alpha$ expression in muscle, we did observe greater IL-6 expression following 2 weeks of bed rest in the control but not the blood flow restricted leg (Fig. 6*G*). These findings agree with previous work, showing an increase in IL-6, but not TNF-$\alpha$, mRNA expression after 7 days of bed rest (Drummond et al., 2013). So far, it remains unknown if blood flow restriction may attenuate the rise in inflammation during disuse and whether this has any clinical relevance. For intracellular amino acid transport (Fig. 6*H–K*), we did not observe any differences between legs over time (only a tendency for SNAT2 to be reduced following bed rest). We did observe a strong decline in muscle GLUT4 mRNA expression in both legs following bed rest, which supports findings (in both GLUT4 protein content and mRNA expression) from previous bed rest studies (Bienso et al., 2012; Dirks, Stephens, et al., 2018; Tabata et al., 1999). The latter agrees with the observation of a progressive decline in whole-body insulin sensitivity (fasting insulin concentrations and HOMA-IR) and glucose tolerance (OGTT) throughout the 2 weeks of bed rest (Figs 8 and 9).

Apart from changes in skeletal muscle mass, strength and protein turnover, a prolonged period ($\geq$7 days) of disuse has also been shown to reduce skeletal muscle oxidative capacity (Abadi et al., 2009; Dirks et al., 2016). Here, we assessed muscle tissue citrate synthase and cytochrome C oxidase activity as markers of mitochondrial health. Skeletal muscle oxidative capacity decreased substantially following 2 weeks of bed rest (Fig. 7), again with no differences between the blood flow restricted and control leg. The latter seems to be at odds with previous work reporting enhanced local muscle oxidative capacity after 7 days of blood flow restriction (Jeffries et al., 2018),

which could be explained by the fact that participants were assessed whilst maintaining their habitual physical activity. Our combined data seem to agree that bed rest lowers mitochondrial content as well as mitochondrial oxidative phosphorylation capacity and that this decline is not attenuated by frequent daily application of blood flow restriction.

In the present study we observed no benefits of frequent daily blood flow restriction on muscle protein synthesis, muscle mass, muscle quality or strength throughout 2 weeks of bed rest. Therefore, based on our current and previous acute findings (Nyakayiru et al., 2019), we would argue that without any muscle contractile stimulus, blood flow restriction does not increase muscle protein synthesis rates or preserve muscle mass during bed rest. There is a possibility that the contrasting findings of the present study in relation to previous studies (Kakehi et al., 2020; Kubota et al., 2008, 2011; Takarada et al., 2000) may be explained by the different disuse model applied (i.e. bed rest *vs.* leg immobilization). However, another recent leg immobilization study also failed to observe any benefit of blood flow restriction on measures of muscle mass and strength (Slysz et al., 2020). It could also be argued that (small) differences in blood flow restriction protocols may be responsible for the observed differences between studies and that the applied protocol in our present study may not have been sufficient to stimulate muscle protein synthesis or inhibit proteolysis. However, this seems unlikely given that we applied greater compressive force during blood flow restriction when compared to previous work (Kubota et al., 2011). Based on our findings we can only conclude that blood flow restriction does not preserve the loss of muscle quantity, quality or muscle function during prolonged bedrest. Considering the health burden of even short periods of forced bedrest (Allen et al., 1999; Asher, 1947; Parry & Puthucheary, 2015), other strategies need to be developed to effectively preserve muscle health during a period of disuse. Active or passive muscle activation, potentially in combination with blood flow restriction, may prove effective in such situations (Barbalho et al., 2019; Dirks, Wall, et al., 2018; Nunes et al., 2022; Reidy et al., 2017; Slysz et al., 2020).

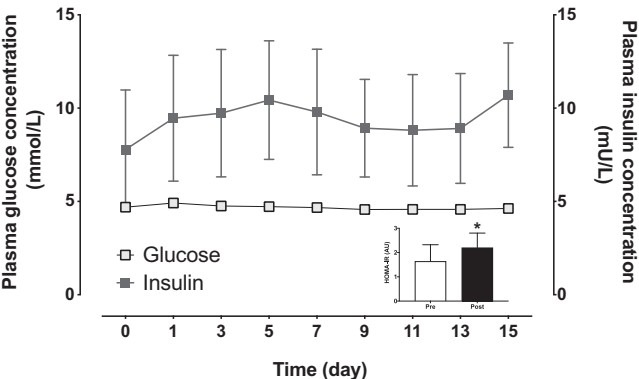

**Figure 9. Plasma glucose and insulin concentrations over time before and during 14 days of strict bed rest and HOMA-IR (insert) before (Pre) and at the end (Post) of 14 days of strict bed rest in healthy, young men (*n* = 12)**
Values represent means $\pm$ 95% CI. *Significantly different ($P$ = 0.023) from pre.

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

## Additional information

### Data availability statement

All data generated and analysed during this study are available from the corresponding author on reasonable request.

### Competing interests

All authors declare no conflicts of interest related to this study.

### Author contributions

C.J.F., L.B.V. and L.J.C.L. conceived and designed the research. C.J.F., W.J.H.H., J.N., M.E.G.W. and J.S.J.S. generated and collected data. C.J.F., W.J.H.H., J.N., M.E.G.W., T.A., J.M.S., T.S., L.B.V. and L.J.C.L. analysed the data and/or interpreted the results. C.J.F. prepared the figures. C.J.F. and L.J.C.L. wrote the manuscript. C.J.F., W.J.H.H., J.N., M.E.G.W., J.S.J.S., T.A., J.M.S., W.K.H.W.W., T.S., L.B.V. and L.J.C.L. revised the manuscript and approved the final version submitted for publication.

### Funding

For this study no external funding was received.

### Acknowledgements

The authors would like to acknowledge Marlou L. Dirks and Andrew M. Holwerda for initial discussions, Wouter Nijsen, Tobias Lampert, Marvin Feldmann, Francesca Badiali, Lisanne Houben, Wendy Sluijsmans, Hasibe Aydeniz, Antoine H. Zorenc, Annemie P. Gijsen and Joy P. B. Goessens for their (technical) assistance, and the enthusiastic support of the subjects who volunteered to participate in this experiment.

### Keywords

disuse, glycaemic control, insulin resistance, muscle fat infiltration, muscle fibre CSA, muscle protein synthesis, nitrogen balance

## Supporting information

Additional supporting information can be found online in the Supporting Information section at the end of the HTML view of the article. Supporting information files available:

**Peer Review History**

