## [Peer Review History · The Journal of Physiology]

Daily blood flow restriction does not preserve muscle mass and strength during 2 weeks of bed rest

Cas J Fuchs, Wesley JH Hermans, Jean Nyakayiru, Michelle EG Weijzen, Joey S.J. Smeets, Thorben Aussieker, Joan Senden, Will KHW Wodzig, Tim Snijders, Lex Verdijk, and Luc JC van Loon

DOI: 10.1113/JP286065

Corresponding author(s): Luc van Loon (l.vanloon@maastrichtuniversity.nl)

The following individual(s) involved in review of this submission have agreed to reveal their identity: Leigh Breen (Referee #1)

Review Timeline:

Submission Date:	29-Nov-2023
Editorial Decision:	04-Jan-2024
Revision Received:	19-Jan-2024
Editorial Decision:	07-Feb-2024
Revision Received:	07-Feb-2024
Accepted:	08-Feb-2024

Senior Editor: Paul Greenhaff

Reviewing Editor: Matthew Brook

Transaction Report:

Dear Professor van Loon,

Re: JP-RP-2023-286065 "Daily blood flow restriction does not preserve muscle mass and strength during 2 weeks of bed rest" by Cas J Fuchs, Wesley JH Hermans, Jean Nyakayiru, Michelle EG Weijzen, Joey S.J. Smeets, Thorben Aussieker, Joan Senden, Will KHW Wodzig, Tim Snijders, Lex Verdijk, and Luc JC van Loon

Thank you for submitting your manuscript to The Journal of Physiology. It has been assessed by a Reviewing Editor and by 2 expert referees and we are pleased to tell you that it is acceptable for publication following satisfactory revision.

REVISION CHECKLIST:

Please upload two versions of your manuscript text: one with all relevant changes highlighted and one clean version with no changes tracked. The manuscript file should include all tables and figure legends, but each figure/graph should be uploaded as separate, high-resolution files. The journal is now integrated with Wiley's Image Checking service. For further details, see: <https://www.wiley.com/en-us/network/publishing/research-publishing/trending-stories/upholding-image-integrity-wileys-image-screening-service>.

- 'Potential Cover Art' for consideration as the issue's cover image
- Appropriate Supporting Information (video, audio or data set: see https://jp.msubmit.net/cgi-bin/main.plex?form_type=display_requirements#supp)

We look forward to receiving your revised submission.

Yours sincerely,

Paul Greenhaff
Senior Editor
The Journal of Physiology

REQUIRED ITEMS

- Author photo and profile. First or joint first authors are asked to provide a short biography (no more than 100 words for one author or 150 words in total for joint first authors) and a portrait photograph. These should be uploaded and clearly labelled together in a Word document with the revised version of the manuscript. See Information for Authors for further details.
- The reference list must be in alphabetical order, rather than numbered, to comply with our Journal format.
- Your manuscript must include a complete Additional Information section, including competing interests; funding; author contributions and acknowledgements.
- Please upload separate high-quality figure files via the submission form.
- You must upload original, uncropped western blot/gel images (including controls) if they are not included in the manuscript. This is to confirm that no inappropriate, unethical or misleading image manipulation has occurred. These should be uploaded as 'Supporting information for review process only'. Please label/highlight the original gels so that we can clearly see which sections/lanes have been used in the manuscript figures. For more information, see: <https://physoc.onlinelibrary.wiley.com/hub/journal-policies#imagmanip>.
- Your paper contains Supporting Information of a type that we no longer publish, including supplementary tables and figures. Any information essential to an understanding of the paper must be included as part of the main manuscript and figures. The only Supporting Information that we publish are video and audio, 3D structures, program codes and large data files. Your revised paper will be returned to you if it does not adhere to our Supporting Information Guidelines.
- Please include an Abstract Figure file, as well as the Figure Legend text within the main article file. The Abstract Figure is a piece of artwork designed to give readers an immediate understanding of the research and should summarise the main conclusions. If possible, the image should be easily 'readable' from left to right or top to bottom. It should show the physiological relevance of the manuscript so readers can assess the importance and content of its findings. Abstract Figures should not merely recapitulate other figures in the manuscript. Please try to keep the diagram as simple as possible and without superfluous information that may distract from the main conclusion(s). Abstract Figures must be provided by authors no later than the revised manuscript stage and should be uploaded as a separate file during online submission labelled as File Type 'Abstract Figure'. Please also ensure that you include the figure legend in the main article file. All Abstract Figures should be created using BioRender. Authors should use The Journal's premium BioRender account to export high-resolution images. Details on how to use and access the premium account are included as part of this email.

EDITOR COMMENTS

Reviewing Editor:

Thank you for submitting to The Journal of Physiology, your manuscript has been reviewed by two expert referees. The referees highlighted the manuscript provides a robust investigation into the effects of blood flow restriction on muscle during 2 weeks of bed rest. The authors have completed a well-controlled study that includes measure of muscle mass, muscle strength, and muscle protein synthesis along with intramuscular signalling and gene expression. The referees have raised some comments throughout the manuscript that need to be fully addressed. This includes some further discussion on how the FSR relate to other work in the field, greater consideration on the wider impacts within muscle, increased rationale for the use of BFR, and others.

In the key points the authors state 'Blood flow restriction applied during bed rest does not increase daily muscle protein synthesis rates and does not preserve muscle mass or strength'. This could be rephrased, as it has not increased, but also not prevented the decline in MPS. I also would discourage the authors from overly strong statements such as 'blood flow restriction does not have any impact on muscle metabolism' as this is only true for the measurements included.

Senior Editor:

Thank you for your manuscript submission to The Journal of Physiology. It has been considered by a reviewing editor and two expert reviewers. All are of the opinion that the manuscript would be of interest to the Journal of Physiology readership and could be impactful. The Reviewing Editor and Reviewer no. 1 have raised some points that the authors should address when revising the manuscript. Thank you for considering the Journal of Physiology and we look forward to receiving the revised version.

REFEREE COMMENTS

Referee #1:

The authors present a rigorous investigation of the impact of thrice daily unilateral BFR on MPS, muscle mass, and strength during 2 weeks of strict bed rest in healthy younger adults. Bed rest significantly reduced muscle mass and strength, whilst BFR during bed rest did not alter daily MPS rates or preserve muscle mass or strength. The study provides comprehensive mechanistic evidence that a robust BFR protocol does not protect muscle health during bed-rest, and addresses key points that underlie the discrepant evidence from previous BFR-disuse studies. The authors are to be commended for completing a challenging experimental study, and presenting a well-written manuscript with a clearly communicated message that will be of great interest to muscle physiologists, rehabilitation specialists and clinicians. I have a few minor points that the authors may wish to consider to strengthen the final manuscript.

- Without a comparison of pre bed-rest FSR for control and BFR legs, it is difficult to understand precisely how immobilization impacted MPS rates in each leg. Rates $<1.1\%/day$ seem lower than previously observed by this group which makes sense after 2 weeks of unloading. It is logical that deficits in MPS were evident in both legs and that this is driving the observed atrophy. Having said that, the authors mitigate this issue linking to evidence from their lab disuse studies or elsewhere to reassure that MPS was likely blunted.

- Similar to the comment directly above relating to no pre bed-rest FSR values, although any potential between leg variability in FSR would likely be very minor (not enough to mask any influence of BFR on FSR) it may be worth stating how FSR values between legs do not differ between dominant and non-dominant.

- Introduction: The strength of using BFR to mitigate disuse muscle atrophy and strength loss is partly underpinned by the negative impact of disuse on microvascular function, capillarization, oxygen/nutrient delivery, as shown in some previous studies. I appreciate the authors did not measure these parameters (adds time, expense to an already complex study) but feel the study rationale would be strengthened by mentioning such impairments as a physiological response to immobilization that BFR would target.

- Line 52: States that no studies have determined the effects of daily passive BFR on muscle mass and strength during immobilization. This statement somewhat contradicts the preceding sentences describing the outcomes of BFR interventions on muscle mass and strength during disuse. If one aspect of the novelty is use of 'passive' BFR or some other aspect of the present BFR protocol, then perhaps add a some specifics of the BFR interventions used in earlier studies to highlight this.

- Line 60: Is there something specific that led the authors to hypothesize that BFR would attenuate muscle atrophy in their hands? From the above introduction text it seems that the findings on BFR during immobilization are contradictory, but the reasons for this are not exactly clear. Highlighting this, and the novelty of the current BFR protocol and study design would strengthen the importance of this work.

- Introduction: There is no mention of the impact of immobilization on mitochondrial content/activity or insulin sensitivity nor how BFR may impact these parameters. Yet these are important secondary outcomes from the study that are discussed extensively later in the manuscript. I think the rationale for including these measures needs to come through clearer from the outset of the introduction. This should also feed into the aims and hypotheses as there is a slight discord with the opening paragraph of the discussion which mentions insulin sensitivity and mitochondrial dysfunction as key outcomes.

- Line 254: great that there was a return visit to measure strength recovery. Is it possible to report the strength values at 6 weeks after bed-rest in the strength results section ? If not then may be best to reove this detail from the methods.

- BFR protocol: I may have missed it, but what were the specific timings of each daily BFR stimulus and was this intended to fall in close proximity to meal intake to maximize the anabolic response?

- Results: Consistency with reporting p values. At times they are specific and other times reported as < or >.

- Line 459: how would the gradual increase in saliva deuterium enrichment affect calculated FSR values, if at all?

- Western blot and gene data: Do differences in gene and protein expression become apparent if data are expressed relative to the pre bed-rest control leg (e.g., normalizing this value to 1.0 and presenting other data pints as fold change)?

- Line 565. In this sentence it may be helpful to include a suggestion of possible discrepancies between previous studies. In thw following sentence this would then allow the authors to define exactly how their study extends on the earlier work.

- Line 579. I assume the authors are referring to MPS rates of 1.6-1.7%/day in younger adults as per the current cohort?

- Line 588: This conclusion makes sense based on previous work from this group showing no effect of BFR on acutely measured myofibrillar protein synthesis. Citing this evidence here would suport this statement (unless the authors have cted elsewhere later on).

- Line 626: "markers of skeletal muscle health" is a little vague, I would change to mitochondrial health, oxidative function, or something similarly reflective.

- Line 653: This final short paragraph isn't required, as the authors state their conclusions at the end of the preceding

paragraph.

Referee #2:

The present manuscript describes a "small" human intervention study investigating the effects of thrice daily in healthy young men. Deuterium derived measures of MPS, anabolic signaling, Nitrogen balance, muscle anabolic signaling, and relevant inflammatory measures were assessed. This group has STRONG expertise in muscle physiology and the measurements employed as such I have little worry about the robustness of the methodology and research. These types of studies are not easy to perform despite the "low n size." I think this manuscript represents high quality work and should be accepted for publication.

Impact on the area of research: Great work, highly controlled and documented. A lot of data and strong methodology. The use of actual bedrest is a strong study design point. The only knock I would have is the use of healthy young men, that said there is still strong clinical utility.

Insight into physiological mechanisms in this field: I think this is the best part of the paper. The authors successfully replicated previous findings and provided additional insights into bedrest induced muscle and strength loss. This manuscript provides a lot of data valuable data showing that in the absence of muscle contraction BFR is not effective at attenuating muscle atrophy. For this reason I expect it will be highly cited within the field. The measurement of D-dimer is a welcome addition and often overlooked in the field of BFR. Every physician I have worked with shares this concern. Having this data out will be helpful for future clinical applications.

Originality of the research: The use of BFR in the absence of muscle contraction in a pseudo-clinical scenario builds upon previous research in the area but is also original in the sense of many bedrest situations don't allow for muscle contraction to occur.

Study design and robustness of the experimental data: Group are world leaders in this field. Have no doubts about this area. Preregistration of the study provides additional confidence in this work.

Validity of the conclusions: The authors conclusions are supported by their data, and as previously mentioned they have replicated previous finding. No complaints here.

END OF COMMENTS

Confidential Review

29-Nov-2023

Response to Referees

REQUIRED ITEMS

- Author photo and profile. First or joint first authors are asked to provide a short biography (no more than 100 words for one author or 150 words in total for joint first authors) and a portrait photograph. These should be uploaded and clearly labelled together in a Word document with the revised version of the manuscript. See Information for Authors for further details.

This has been added.

- The reference list must be in alphabetical order, rather than numbered, to comply with our Journal format.

Revised accordingly.

- Your manuscript must include a complete Additional Information section, including competing interests; funding; author contributions and acknowledgements.

This has been included according to the guidelines.

- Please upload separate high-quality figure files via the submission form.

All individual figures have been uploaded.

- You must upload original, uncropped western blot/gel images (including controls) if they are not included in the manuscript. This is to confirm that no inappropriate, unethical or misleading image manipulation has occurred. These should be uploaded as 'Supporting information for review process only'. Please label/highlight the original gels so that we can clearly see which sections/lanes have been used in the manuscript figures. For more information, see: <https://physoc.onlinelibrary.wiley.com/hub/journal-policies#imagmanip>.

We have included original western blot images into the main manuscript (see Figure 5).

We have now also submitted and uploaded the original gels including the highlighted lanes that are used in the manuscript, as Supporting information for review process only.

- Your paper contains Supporting Information of a type that we no longer publish, including supplementary tables and figures. Any information essential to an understanding of the paper must be included as part of the main manuscript and figures. The only Supporting Information that we publish are video and audio, 3D structures, program codes and large data files. Your revised paper will be returned to you if it does not adhere to our Supporting Information Guidelines.

We have now included these figures into the main manuscript.

- Please include an Abstract Figure file, as well as the Figure Legend text within the main article file. The Abstract Figure is a piece of artwork designed to give readers an immediate understanding of the research and should summarise the main conclusions. If possible, the image should be easily 'readable' from left to right or top to bottom. It should show the physiological relevance of the manuscript so readers can assess the importance and content of its findings. Abstract Figures should not merely recapitulate other figures in the manuscript. Please try to keep the diagram as simple as

possible and without superfluous information that may distract from the main conclusion(s). Abstract Figures must be provided by authors no later than the revised manuscript stage and should be uploaded as a separate file during online submission labelled as File Type 'Abstract Figure'. Please also ensure that you include the figure legend in the main article file. All Abstract Figures should be created using BioRender. Authors should use The Journal's premium BioRender account to export high-resolution images. Details on how to use and access the premium account are included as part of this email.

We have added an Abstract Figure including legend in the manuscript file.

EDITOR COMMENTS

Reviewing Editor:

Thank you for submitting to The Journal of Physiology, your manuscript has been reviewed by two expert referees. The referees highlighted the manuscript provides a robust investigation into the effects of blood flow restriction on muscle during 2 weeks of bed rest. The authors have completed a well-controlled study that includes measure of muscle mass, muscle strength, and muscle protein synthesis along with intramuscular signalling and gene expression. The referees have raised some comments throughout the manuscript that need to be fully addressed. This includes some further discussion on how the FSR relate to other work in the field, greater consideration on the wider impacts within muscle, increased rationale for the use of BFR, and others.

In the key points the authors state 'Blood flow restriction applied during bed rest does not increase daily muscle protein synthesis rates and does not preserve muscle mass or strength'. This could be rephrased, as it has not increased, but also not prevented the decline in MPS. I also would discourage the authors from overly strong statements such as 'blood flow restriction does not have any impact on muscle metabolism' as this is only true for the measurements included.

We would like to thank the Reviewing Editor for the time to read and provide feedback on our manuscript. We have revised the manuscript accordingly. We have made adjustments based on the comments provided, which can be found in **green in the manuscript file.**

For the specific comments raised by the reviewing editor, see revisions:

- **Third key point and final sentence in the abstract (line: 18).**
- **Lines: 676-679 ("Therefore, based on our current and previous acute findings (Nyakayiru *et al.*, 2019), we would argue that without any muscle contractile stimulus, blood flow restriction does not increase muscle protein synthesis rates or preserve muscle mass during bed rest ~~have any impact on muscle metabolism.~~").**

Senior Editor:

Thank you for your manuscript submission to The Journal of Physiology. It has been considered by a reviewing editor and two expert reviewers. All are of the opinion that the manuscript would be of interest to the Journal of Physiology readership and could be impactful. The Reviewing Editor and Reviewer no. 1 have raised some points that the authors should address when revising the manuscript. Thank you for considering the Journal of Physiology and we look forward to receiving the revised version.

REFeree COMMENTS

Referee #1:

The authors present a rigorous investigation of the impact of thrice daily unilateral BFR on MPS, muscle mass, and strength during 2 weeks of strict bed rest in healthy younger adults. Bed rest significantly reduced muscle mass and strength, whilst BFR during bed rest did not alter daily MPS rates or preserve muscle mass or strength. The study provides comprehensive mechanistic evidence that a robust BFR protocol does not protect muscle health during bed-rest, and addresses key points that underlie the discrepant evidence from previous BFR-disuse studies. The authors are to be commended for completing a challenging experimental study, and presenting a well-written manuscript with a clearly communicated message that will be of great interest to muscle physiologists, rehabilitation specialists and clinicians. I have a few minor points that the authors may wish to consider to strengthen the final manuscript.

We would like to thank reviewer 1 for the time to read and provide feedback on our manuscript and appreciate the kind words. We have revised the manuscript accordingly. All adjustments can be found in green in the manuscript file.

- Without a comparison of pre bed-rest FSR for control and BFR legs, it is difficult to understand precisely how immobilization impacted MPS rates in each leg. Rates $<1.1\%/day$ seem lower than previously observed by this group which makes sense after 2 weeks of unloading. It is logical that deficits in MPS were evident in both legs and that this is driving the observed atrophy. Having said that, the authors mitigate this issue linking to evidence from their labs disuse studies or elsewhere to reassure that MPS was likely blunted.

We agree with this reviewer. The main comparison was if there were differences in daily MPS between a leg subjected to BFR vs a CON leg during a prolonged period of bed rest. Here we found no differences. Hence, we can conclude that BFR treatment does not impact daily MPS during bed rest.

We did not assess habitual MPS rates (before the bed rest period) as this was not an objective in the present study. However, in order to still provide an insight into a general effect of disuse on MPS, we put our findings into perspective based on previous work and discussed this in the discussion section.

- Similar to the comment directly above relating to no pre bed-rest FSR values, although any potential between leg variability in FSR would likely be very minor (not enough to mask any influence of BFR on FSR) it may be worth stating how FSR values between legs do not differ between dominant and non-dominant.

We also calculated the FSR between the dominant (Mean \pm SD: $1.09\pm 0.12\%/day$) and non-dominant ($1.10\pm 0.14\%/day$) legs. Here we also did not observe any difference between the legs ($P=0.767$). We included this in the result section to inform the reader about these data.

See lines: 483-485 ("Furthermore, no differences were observed in mixed muscle protein synthesis rates between the dominant or non-dominant leg (1.09 ± 0.12 vs $1.10\pm 0.14\% \cdot d^{-1}$, respectively; $P=0.767$).").

- Introduction: The strength of using BFR to mitigate disuse muscle atrophy and strength loss is partly underpinned by the negative impact of disuse on microvascular function, capillarization, oxygen/nutrient delivery, as shown in some previous studies. I appreciate the authors did not measure these parameters (adds time, expense to an already complex study) but feel the study rationale would be strengthened by mentioning such impairments as a physiological response to immobilization that BFR would target.

We have added a sentence in the introduction to include the potential of BFR to stimulate angiogenesis. See lines: 44-45 (“This may stimulate angiogenesis and augment muscle oxidative capacity (Jeffries *et al.*, 2018; Pignanelli *et al.*, 2021; Li *et al.*, 2022).”).

- Line 52: States that no studies have determined the effects of daily passive BFR on muscle mass and strength during immobilization. This statement somewhat contradicts the preceding sentences describing the outcomes of BFR interventions on muscle mass and strength during disuse. If one aspect of the novelty is use of 'passive' BFR or some other aspect of the present BFR protocol, then perhaps add a some specifics of the BFR interventions used in earlier studies to highlight this.

We particularly focused here on the impact of BFR on ‘bed rest’ as the disuse model (which is novel), rather than limb immobilization. In order to make this clearer for the reader we have changed the sentence.

See line: 61-64 (“Whereas previous studies have utilized limb immobilization as the model of disuse ~~So far~~, no studies have investigated the effects of daily passive blood flow restriction on skeletal muscle mass and strength during a prolonged period of bed rest ~~on skeletal muscle mass and strength.~~”).

- Line 60: Is there something specific that led the authors to hypothesize that BFR would attenuate muscle atrophy in their hands? From the above introduction text it seems that the findings on BFR during immobilization are contradictory, but the reasons for this are not exactly clear. Highlighting this, and the novelty of the current BFR protocol and study design would strengthen the importance of this work.

Given that most studies have shown a beneficial effect of BFR in preserving muscle mass and strength during limb immobilization, we speculated that this would also occur during a prolonged period of bed rest. We have added this in the introduction to make it clearer for the readers.

See lines: 70-75 (“Given that many studies have shown blood flow restriction to mitigate the decline in both muscle quantity and quality during limb immobilization, we ~~We~~ hypothesized that daily blood flow restriction would attenuate the decline in muscle oxidative capacity, activate anabolic signaling pathways, increase muscle protein synthesis rates and, as such, attenuate the loss of skeletal muscle mass and strength during 2 weeks of bed rest.”).

- Introduction: There is no mention of the impact of immobilization on mitochondrial content/activity or insulin sensitivity nor how BFR may impact these parameters. Yet these are important secondary outcomes from the study that are discussed extensively later in the manuscript. I think the rationale for including these measures needs to come through clearer from the outset of the introduction. This should also feed into the aims and hypotheses as there is a slight discord with the opening paragraph of the discussion which mentions insulin sensitivity and mitochondrial dysfunction as key outcomes.

We have added this in the introduction to include the reported impact of bed rest on muscle

oxidative capacity, and also the potential of BFR to enhance muscle oxidative capacity, which now also feeds into our hypothesis. Overall, it now aligns better with the data that we report.

See lines: 27-31 ("This reduces insulin sensitivity, decreases muscle oxidative capacity, lowers muscle protein synthesis rates, and leads to a progressive loss of skeletal muscle mass and strength in humans of all ages (Stuart *et al.*, 1988; Wall & van Loon, 2013; Dirks *et al.*, 2016; English *et al.*, 2016).").

Lines: 44-45 ("This may stimulate angiogenesis and augment muscle oxidative capacity (Jeffries *et al.*, 2018; Pignanelli *et al.*, 2021; Li *et al.*, 2022).").

And lines: 72-75 ("~~we~~ We hypothesized that daily blood flow restriction would attenuate the decline in muscle oxidative capacity, activate anabolic signaling pathways, increase muscle protein synthesis rates and, as such, attenuate the loss of skeletal muscle mass and strength during 2 weeks of bed rest.").

We reported the known impact of bed rest on insulin sensitivity in the second sentence of the introduction (see lines: 27-31) and we also provide our own data later in the manuscript (Figure 8 & 9). We did not add this into the hypothesis as the current design only allowed us to investigate the impact of bed rest on whole-body insulin sensitivity and it did not allow us to look specifically at the impact of BFR on insulin sensitivity.

- Line 254: great that there was a return visit to measure strength recovery. Is it possible to report the strength values at 6 weeks after bed-rest in the strength results section? If not then may be best to remove this detail from the methods.

We indeed have reported this in the result section. Please see lines: 469-472

("After ~6 weeks, 1RM leg press and extension strength were not different from 1RM values obtained prior to bed rest in both the CON and BFR leg (leg press: 153±37 kg in CON, 147±43 kg in BFR ($P=0.700$); leg extension: 62±14 kg in CON, 61±13 kg in BFR ($P=0.105$)).").

- BFR protocol: I may have missed it, but what were the specific timings of each daily BFR stimulus and was this intended to fall in close proximity to meal intake to maximize the anabolic response?

We applied the BFR protocol in between meal moments at ~11am, ~3pm, ~8pm.

See lines: 102-105 ("During the 14-day bed rest period, food was provided regularly (breakfast, lunch, dinner, and snacks in between) and during 3 times a day (~11am; ~3pm; ~8pm) unilateral blood flow restriction (BFR) was applied.")

And lines: 212-214 ("Blood flow restriction was applied daily for 3 times a day (at ~11am, ~3pm, and ~8pm) with a lower extremity pressure cuff (SC10, Hokanson, Bellevue, WA) that was placed around the most proximal portion of each thigh while the subject was lying on bed.").

We did not necessarily intend to apply the BFR protocol in close proximity to meal intake, but rather in between the meal moments. It is also hard to tell what the best timing would be, given that protein digestion and amino acid absorption following mixed meal intake will take several hours to reach its peak in plasma amino acid concentrations (e.g., PMID: 37972895). So, we intended to spread it over the day and see if there would be a general anabolic effect of BFR treatment (whether due to a synergistic effect with meal intake or not).

- Results: Consistency with reporting p values. At times they are specific and other times reported as < or >.

We have indeed done this where the P value is lower than $P < 0.001$ or if we combined more data in one sentence and state the significance of multiple comparisons ($P < 0.05$). For the other data we ensured to provide the actual P value.

- Line 459: how would the gradual increase in saliva deuterium enrichment affect calculated FSR values, if at all?

This should have no impact on the FSR values/comparison as we utilized a within-subject design (i.e., both the BFR and CON leg had the same precursor pool).

- Western blot and gene data: Do differences in gene and protein expression become apparent if data are expressed relative to the pre bed-rest control leg (e.g., normalizing this value to 1.0 and presenting other data points as fold change)?

We indeed also checked our data when expressed as fold change. However, this did not change the conclusions. As such, we decided to provide the mean data (including the individual data points) as arbitrary units in the manuscript to ensure the reader would get most insight into the data as we analyzed it.

- Line 565. In this sentence it may be helpful to include a suggestion of possible discrepancies between previous studies. In the following sentence this would then allow the authors to define exactly how their study extends on the earlier work.

Although speculative, a potential discrepancy could be explained by variations in observed rates of muscle loss across previous immobilization and BFR studies, where BFR could become more effective with higher rates of disuse-induced atrophy. Another reason could be the applied frequency or volume of the BFR treatment (both aspects have also been discussed before, PMID: 33105390). In line with the comment of this reviewer, we added a sentence to clarify this for the reader.

See lines: 596-599 ("It has been suggested that the discrepancy between studies could be due to blood flow restriction being more effective in scenarios involving more severe atrophy or wasting and/or the applied frequency or volume of the blood flow restriction treatment (Slysz *et al.*, 2020).").

- Line 579. I assume the authors are referring to MPS rates of 1.6-1.7%/day in younger adults as per the current cohort?

This is correct, we have added this information to make this clearer for the readers.

See lines: 612-613 ("Previously, we have shown that under normal resting conditions (also in healthy young males), muscle protein synthesis rates average around ~1.6-1.7 %/d (Holwerda *et al.*, 2018).").

- Line 588: This conclusion makes sense based on previous work from this group showing no effect of BFR on acutely measured myofibrillar protein synthesis. Citing this evidence here would support this statement (unless the authors have cited elsewhere later on).

We agree with this. Therefore, we have cited this work in the preceding sentence to make this point clear.

See lines: 619-624 ("The absence of a stimulating effect of three daily sessions of blood flow restriction during prolonged bed rest seems to be in line with our inability to detect an increase in muscle protein synthesis rates following acute recovery from a single session of blood flow restriction (Nyakayiru *et al.*, 2019). Consequently, we must conclude that a single session or more

prolonged daily application of blood flow restriction do not increase muscle protein synthesis rates.”).

- Line 626: "markers of skeletal muscle health" is a little vague, I would change to mitochondrial health, oxidative function, or something similarly reflective.

Revised accordingly. Please see lines: 664-666 (“Here, we assessed muscle tissue citrate synthase and cytochrome C oxidase activity as markers of skeletal muscle health mitochondrial health.”).

- Line 653: This final short paragraph isn't required, as the authors state their conclusions at the end of the preceding paragraph.

We have removed these two sentences.

Referee #2:

The present manuscript describes a "small" human intervention study investigating the effects of thrice daily in healthy young men. Deuterium derived measures of MPS, anabolic signaling, Nitrogen balance, muscle anabolic signaling, and relevant inflammatory measures were assessed. This group has STRONG expertise in muscle physiology and the measurements employed as such I have little worry about the robustness of the methodology and research. These types of studies are not easy to perform despite the "low n size." I think this manuscript represents high quality work and should be accepted for publication.

Impact on the area of research: Great work, highly controlled and documented. A lot of data and strong methodology. The use of actual bedrest is a strong study design point. The only knock I would have is the use of healthy young men, that said there is still strong clinical utility.

Insight into physiological mechanisms in this field: I think this is the best part of the paper. The authors successfully replicated previous findings and provided additional insights into bedrest induced muscle and strength loss. This manuscript provides a lot of data valuable data showing that in the absence of muscle contraction BFR is not effective at attenuating muscle atrophy. For this reason I expect it will be highly cited within the field. The measurement of D-dimer is a welcome addition and often overlooked in the field of BFR. Every physician I have worked with shares this concern. Having this data out will be helpful for future clinical applications.

Originality of the research: The use of BFR in the absence of muscle contraction in a pseudo-clinical scenario builds upon previous research in the area but is also original in the sense of many bedrest situations don't allow for muscle contraction to occur.

Study design and robustness of the experimental data: Group are world leaders in this field. Have no doubts about this area. Preregistration of the study provides additional confidence in this work.

Validity of the conclusions: The authors conclusions are supported by their data, and as previously mentioned they have replicated previous finding. No complaints here.

We would like to thank reviewer 2 for the time to read and provide feedback on our manuscript and appreciate the kind words.

Dear Professor van Loon,

Re: JP-RP-2024-286065R1 "Daily blood flow restriction does not preserve muscle mass and strength during 2 weeks of bed rest" by Cas J Fuchs, Wesley JH Hermans, Jean Nyakayiru, Michelle EG Weijzen, Joey S.J. Smeets, Thorben Aussieker, Joan Senden, Will KHW Wodzig, Tim Snijders, Lex Verdijk, and Luc JC van Loon

Thank you for submitting your manuscript to The Journal of Physiology. It has been assessed by a Reviewing Editor and by 1 expert referee and we are pleased to tell you that it is almost ready for acceptance. Before formal acceptance, however, we would be grateful if you could attend to the minor comments of the Editor regarding Figures 1-7.

The review reports are copied at the end of this email.

REVISION CHECKLIST:

Please upload two versions of your manuscript text: one with all relevant changes highlighted and one clean version with no changes tracked. The manuscript file should include all tables and figure legends, but each figure/graph should be uploaded as separate, high-resolution files. The journal is now integrated with Wiley's Image Checking service. For further details, see: <https://www.wiley.com/en-us/network/publishing/research-publishing/trending-stories/upholding-image-integrity-wileys-image-screening-service>.

- 'Potential Cover Art' for consideration as the issue's cover image
- Appropriate Supporting Information (video, audio or data set: see https://jp.msubmit.net/cgi-bin/main.plex?form_type=display_requirements#supp)

We look forward to receiving your revised submission.

Yours sincerely,

Paul Greenhaff
Senior Editor
The Journal of Physiology

REQUIRED ITEMS

Please comply with our statistics policy: https://jp.msubmit.net/cgi-bin/main.plex?form_type=display_requirements#statistics

EDITOR COMMENTS

Reviewing Editor:

The authors have addressed the referee's comments.

Senior Editor:

Thank you for the revised manuscript. The Reviewing Editor and Referee no. 1 are happy with the changes made by the authors. It is deemed acceptable for publication. However, it would be helpful to the readership if Figures 1 to 7 (inclusive) also included the standard deviation of the mean to reflect the the extent of randomness of individuals about their common average. This is not easily conveyed by showing mean and individual values alone.

REFeree COMMENTS

Referee #1:

Thank you for the attention paid to the comments, I have no further suggestions. Congratulations!

END OF COMMENTS

1st Confidential Review

19-Jan-2024

Response to Referees

Reviewing Editor:

The authors have addressed the referee's comments.

Senior Editor:

Thank you for the revised manuscript. The Reviewing Editor and Referee no. 1 are happy with the changes made by the authors. It is deemed acceptable for publication. However, it would be helpful to the readership if Figures 1 to 7 (inclusive) also included the standard deviation of the mean to reflect the the extent of randomness of individuals about their common average. This is not easily conveyed by showing mean and individual values alone.

This has been revised accordingly, both in figures and text.

REFEREE COMMENTS

Referee #1:

Thank you for the attention paid to the comments, I have no further suggestions. Congratulations!

Dear Dr van Loon,

Re: JP-RP-2024-286065R2 "Daily blood flow restriction does not preserve muscle mass and strength during 2 weeks of bed rest" by Cas J Fuchs, Wesley JH Hermans, Jean Nyakayiru, Michelle EG Weijzen, Joey S.J. Smeets, Thorben Aussieker, Joan Senden, Will KHW Wodzig, Tim Snijders, Lex Verdijk, and Luc JC van Loon

We are pleased to tell you that your paper has been accepted for publication in The Journal of Physiology.

Authors should note that it is too late at this point to offer corrections prior to proofing. Major corrections at proof stage, such as changes to figures, will be referred to the Editors for approval before they can be incorporated. Only minor changes, such as to style and consistency, should be made at proof stage. Changes that need to be made after proof stage will usually require a formal correction notice.

Yours sincerely,

Paul Greenhaff
Senior Editor
The Journal of Physiology

P.S. - You can help your research get the attention it deserves! Check out Wiley's free Promotion Guide for best-practice recommendations for promoting your work at www.wileyauthors.com/eeo/guide. You can learn more about Wiley Editing Services which offers professional video, design, and writing services to create shareable video abstracts, infographics, conference posters, lay summaries, and research news stories for your research at www.wileyauthors.com/eeo/promotion.

IMPORTANT NOTICE ABOUT OPEN ACCESS: To assist authors whose funding agencies mandate public access to published research findings sooner than 12 months after publication, The Journal of Physiology allows authors to pay an Open Access (OA) fee to have their papers made freely available immediately on publication.

You can check if your funder or institution has a Wiley Open Access Account here: <https://authorservices.wiley.com/author-resources/Journal-Authors/licensing-and-open-access/open-access/author-compliance-tool.html>.

EDITOR COMMENTS

Thank you for the final amendments. Congratulations. Thank you for considering the Journal of Physiology.